# Machine Learning-Driven Identification of Exosome-Related Genes in Head and Neck Squamous Cell Carcinoma for Prognostic Evaluation and Drug Response Prediction

**DOI:** 10.3390/biomedicines13040780

**Published:** 2025-03-23

**Authors:** Hua Cai, Liuqing Zhou, Yao Hu, Tao Zhou

**Affiliations:** 1Department of Otorhinolaryngology, Union Hospital, Tongji Medical College, Huazhong University of Science and Technology, Wuhan 430022, China; 2020xh0035@hust.edu.cn (H.C.); 2013xh0823@hust.edu.cn (L.Z.); 2Department of Otorhinolaryngology, The Central Hospital of Wuhan, Wuhan 430021, China

**Keywords:** exosome, head and neck squamous cell carcinoma, prognosis, immune infiltration, bioinformatics

## Abstract

**Background:** This study integrated four Gene Expression Omnibus (GEO) datasets to identify disease-specific feature genes in head and neck squamous cell carcinoma (HNSCC) through differential expression analysis with batch effect correction. **Methods**: The GeneCards database was used to find genes related to exosomes, and samples were categorized into groups with high and low expression levels based on these feature genes. Functional and pathway enrichment analyses (GO, KEGG, and GSEA) were used to investigate the possible biological mechanisms underlying feature genes. A predictive model was produced by using machine learning algorithms (LASSO regression, SVM, and random forest) to find disease-specific feature genes. Receiver operating characteristic (ROC) curve analysis was used to assess the model’s effectiveness. The diagnostic model showed excellent predictive accuracy through external data GSE83519 validation. **Results**: This analysis highlighted 22 genes with significant differential expression. A predictive model based on five important genes (*AGRN*, *TSPAN6*, *MMP9*, *HBA1*, and *PFN2*) was produced by using machine learning algorithms. *MMP9* and *TSPAN6* showed relatively high predictive performance. Using the ssGSEA algorithm, three key genes (*MMP9*, *AGRN*, and *PFN2*) were identified as strongly linked to immune regulation, immune response suppression, and critical signaling pathways involved in HNSCC progression. Matching HNSCC feature gene expression profiles with DSigDB compound signatures uncovered potential therapeutic targets. Molecular docking simulations identified ligands with high binding affinity and stability, notably C5 and Hoechst 33258, which were prioritized for further validation and potential drug development. **Conclusions**: This study employs a novel diagnostic model for HNSCC constructed using machine learning technology, which can provide support for the early diagnosis of HNSCC and thus contribute to improving patient treatment plans and clinical management strategies.

## 1. Introduction

Head and neck squamous cell carcinoma (HNSCC) is a difficult malignancy caused by the epithelial lining of the larynx, pharynx, and oral cavity. The most common risk factors are human papillomavirus (HPV) infection, tobacco use, and alcohol consumption. While treatment options such as surgery, radiotherapy, immunotherapy, and chemotherapy have advanced, outcomes for patients with advanced HNSCC remain unfavorable.

Recent studies have focused on exosomes because of their crucial function in the tumor microenvironment, which may affect HNSCC progression, diagnosis, and treatment. These nanoscale extracellular vesicles, measuring 30–100 nm, are stable in blood, serous effusion, urine, saliva, and cerebrospinal fluid, among other bodily fluids [1]. They facilitate intercellular communication by transferring biomolecules such as proteins, lipids, and nucleic acids, thereby impacting processes such as differentiation, cell proliferation, and immune responses [2]. Recent studies have demonstrated the potential of exosomes as diagnostic and prognostic biomarkers by connecting bioactive components such as RNA, DNA, and proteins to the initiation and spread of tumors [3].

In this work, we investigate the pivotal role of exosomes in HNSCC and evaluate their clinical potential as biomarkers and therapeutic targets. Studies have consistently shown that exosomes contribute to key processes such as tumor growth, metastasis, immune evasion, and disease progression [4,5,6]. Acting as mediators between cancer and non-cancerous cells, they transfer crucial molecules that regulate cellular interactions [7]. However, the specific mechanisms by which exosome-mediated gene regulation drives HNSCC progression remain poorly understood. Our research focuses on unraveling the biological functions of exosome-related genes (ERGs) in HNSCC and assessing their prognostic value.

## 2. Materials and Methods

### 2.1. Patient Dataset

Microarray datasets containing differentially expressed mRNAs in normal and tumor tissues of HNSCC patients were sourced from the GEO database, specifically GSE39400, GSE23036, GSE6631, GSE29330, and GSE83519. Furthermore, 6,883 genes related to exosomes were obtained from the GeneCards database (https://www.genecards.org, accessed on 7 January 2025), a user-friendly resource providing comprehensive details on annotated and predicted human genes. From these, 861 protein-coding genes with relevance scores above 2 were filtered for further analysis. The GeneCards data are freely accessible and downloadable for research purposes.

### 2.2. Data Processing and Identification of Differentially Expressed Exosome-Related Genes

Raw expression data for each GEO dataset were downloaded and preprocessed using standard methods. The datasets GSE39400, GSE23036, GSE6631, GSE29330, and GSE83519 were normalized and corrected using the “limma” package in R software (version 4.4.2, https://www.r-project.org). The sva R package (version 3.20.0) ComBat method was used to handle batch effects across datasets GSE39400, GSE23036, GSE6631, and GSE29330. The processed datasets were subsequently combined into a single cohort (MergeCohort) consisting of 43 normal and 126 tumor samples. The dataset GSE83519 was selected as an external validation set, including 22 normal and 22 tumor samples.

The DEGs between HNSCC and healthy control samples were found by analyzing the MergeCohort expression matrix. Genes that satisfied the requirements of fold change (FC) > 1 and an adjusted *p*-value < 0.05 were deemed to be significantly different and identified as target genes associated with HNSCC. Genes with logFC > 0 and a *p*-value < 0.05 were considered upregulated, while those with logFC < 0 and a *p*-value < 0.05 were deemed downregulated.

To visualize the DEGs, the R packages “pheatmap” and “ggplot2” were used. The overall distribution of DEGs was displayed using a volcano plot, and the top 50 upregulated and downregulated genes were highlighted in a heatmap. These DEGs were intersected with ERGs using the R’s “ggvenn” package, producing a list of differentially expressed exosome-related genes (DEEGs) for additional research.

### 2.3. Functional Enrichment (GO) and Pathway Analysis (KEGG)

Functional enrichment and pathway analyses were performed using the “ClusterProfiler” R package (http://bioconductor.org) in order to identify the main biological characteristics of the identified DEEGs [8].

Gene Ontology (GO) [9] analysis focused on exploring relevant biological processes, while the Kyoto Encyclopedia of Genes and Genomes (KEGG) [10] pathway analysis was utilized to pinpoint possible signaling pathways linked to these genes. To ensure reliability, both analyses used a *p*-value of less than 0.05 for significance.

### 2.4. GSEA Results

Gene set enrichment analysis (GSEA) was performed to decipher key pathways linked to the most critical ERGs in HNSCC. The analysis assessed whether specific biological processes showed notable enrichment within the dataset. Enrichment scores were calculated to determine statistical relevance, highlighting the top five KEGG pathways with the most meaningful results. Pathways with adjusted *p*-values below 0.05 were deemed statistically significant.

### 2.5. Construction of a Risk Model for Prognostic HNSCC Exosome-Related Genes

This study investigated the relationship between HNSCC ERG expression and patient prognosis by building a predictive diagnostic model.

The LASSO algorithm integrates regularization and feature selection, enhancing predictive accuracy by penalizing large coefficients and reducing model complexity. Logistic regression was performed with the “glmnet” package, applying 10-fold cross-validation to determine the optimal penalty coefficient (λ). Statistical significance was determined using a *p*-value threshold of <0.05.

The support vector machine (SVM) algorithm, a machine learning method, is effective for both regression and classification tasks. It identifies an optimal hyperplane in the feature space, maximizing separation between classes or minimizing regression errors [11]. The e1071 package was employed to implement the SVM algorithm.

The random forest (RF) algorithm is an ensemble learning technique that constructs multiple decision trees to boost prediction accuracy and reliability. For classification tasks, it uses majority voting among trees, and for regression, it averages their predictions [12]. Using the random forest package in R, a diagnostic model was developed with 500 trees (ntree = 500) and three variables per split (mtry = 3). Based on the average decline in the Gini coefficient, each variable’s significance was ranked, and the top 22 DEGs were arranged in that order.

Cross-validation was performed using the LASSO, SVM, and RF algorithms to minimize the risk of overfitting and identify genes with diagnostic significance. To visualize differences in gene expression between the control and treatment groups, boxplots and violin plots of the target genes were created. This provided a clear comparison of target gene expression across different sample groups.

### 2.6. Receiver Operating Characteristic Curve Analysis and Prognostic Nomograms

Receiver operating characteristic (ROC) curve analysis was used to evaluate the intersection model’s gene diagnostic performance. The *x*-axis displayed the false positive rate (1—specificity), and the *y*-axis displayed the true positive rate (TPR). Predictive accuracy was measured by calculating the area under the curve (AUC) values and their 95% confidence intervals (CIs). Genes with an AUC greater than 0.7 were considered to have a strong prognostic value for HNSCC. A prognostic nomogram was also created to assess the risk of HNSCC using the expression levels of risk genes. The model’s accuracy was evaluated through calibration curves, while decision curve analysis (DCA) further validated its performance in distinguishing between control and experimental group samples. To confirm the effectiveness of our prediction approach, it was tested on the external dataset GSE83519.

### 2.7. ssGSEA and Immune Score Risk Exosome-Related Gene Correlation in HNSCC

Single-sample gene set enrichment analysis (ssGSEA) was applied to explore immune-related characteristics within our study samples. Using 29 immune gene sets, which included genes linked to pathways, checkpoints, and immune cell types, ssGSEA was deployed to quantify the enrichment levels of these sets in each sample. The GSVA, GSEABase, and limma R packages were employed for the analysis, providing a detailed immunological profile for each case [13]. Boxplots were used to compare immune-related groupings and gene expression levels identified through ssGSEA. Correlation analysis between key genes and immune cells was performed with the limma, reshape2, tidyverse, and ggplot2 R packages, helping identify statistically significant relationships between genes and immune cell populations.

### 2.8. Prediction of Candidate Drugs

To explore potential therapeutic applications, protein–drug interactions were analyzed using the Drug Signatures Database (DSigDB; http://dsigdb.tanlab.org/DSigDBv1.0/, accessed on 1 February 2025) [14]. A comprehensive resource links genes to 17,389 unique compounds and 22,527 gene sets, providing a robust basis for predicting drug candidates. Target genes identified in the study were added to DSigDB, allowing for the identification of drugs with potential therapeutic value and facilitating the further evaluation of these targets.

### 2.9. Molecular Docking

Molecular docking simulations were conducted to analyze how ligands interact with their target proteins. This method provided insights into binding affinities and interaction modes, allowing the prioritization of promising drug candidates for experimental validation. Protein structures were sourced from the Protein Data Bank (PDB; http://www.rcsb.org), while ligand information was sourced from the PubChem Compound Database [15]. The CB-Dock2 platform (https://cadd.labshare.cn/cb-dock2/index.php, accessed on 1 February 2025) was utilized for docking, automating the process through steps such as cavity detection, docking, and visualization. Input data included PDB files for proteins and MOL2, SDF, or PDB formats for ligands. CB-Dock2 predicted binding sites and affinities based on the three-dimensional structures of proteins and ligands, aiding in drug discovery. In this study, the top five identified drugs were docked with proteins encoded by their target genes, generating complex structures for further analysis.

### 2.10. Statistical Analyses

Statistical analyses were performed using both R and Perl software (version 4.3.1, packages: sva, limma, pheatmap, survival, survminer, glmnet, timeROC, ggpubr, regplot, rms, ggplot2, ggExtra, reshape2, tidyverse, preprocessCore, RColorBrewer, car, ridge, genefilter, biomaRt, GenomicFeatures, maftools, stringr, org.Hs.eg.db, oncoPredict, and remotes). Unless otherwise specified, all analyses in this study considered an estimated *p*-value of <0.05 as statistically significant. For all comparisons: * *p* < 0.05, ** *p* < 0.01, and *** *p* < 0.001.

A flowchart summarizing the key concepts and steps followed throughout this study is illustrated in Figure 1.

## 3. Results

### 3.1. Identification of Differentially Expressed Exosome-Related Genes in HNSCC

Datasets GSE39400, GSE23036, GSE6631, and GSE29330 collectively included data from 126 HNSCC patients and 43 control samples. To account for batch effects introduced by variations in experimental conditions and platforms, the ComBat function was employed for correction. A total of 8227 genes were found after combining the data from these four datasets. Prior to batch correction, the expression data underwent principal component analysis (PCA), which demonstrated batch effect-based clustering (Figure 2A). In contrast, PCA performed post-normalization demonstrated a uniform distribution across datasets, confirming the successful correction of batch effects and the reliable identification of molecular differences (Figure 2B). The combined datasets, referred to as the MergeCohort, exhibited a uniform distribution between HNSCC samples and controls in the PCA plot (Figure 2B). From the merged expression matrix, 251 genes were identified as differentially expressed (|logFC| > 0.585, *p*-value < 0.05), with 118 upregulated genes and 133 downregulated genes (Figure 2C,D). Additionally, 861 protein-coding ERGs with a relevance score above 2 were retrieved from the GeneCards database. By intersecting these two gene sets, 22 DEEGs were identified, including 9 upregulated and 13 downregulated genes (Figure 2E). The upregulated genes included *AGRN*, *CLIC4*, *FN1*, *MMP9*, *THY1*, *PFN2*, *BST2*, *EPCAM*, and *LRRC15*, while the downregulated ones were *TSPAN6*, *MUC1*, *EXPH5*, *TF*, *PIP*, *LCN2*, *TGFBR3*, *CEACAM5*, *FCGBP*, *CLU*, *LTF*, *HBA1*, and *KRT13*.

### 3.2. Functional Enrichment Analysis of Differentially Expressed Exosome-Related Genes

The biological roles of the identified 22 DEEGs were examined using GO annotation and KEGG pathway enrichment analyses, with particular attention paid to their roles in molecular functions (MFs), cellular components (CCs), and biological processes (BPs) (Figure 3A–F). The GO enrichment analysis highlighted significant associations with processes such as the negative regulation of cell migration, cell motility, and locomotion. For cellular components, the genes were predominantly enriched in structures such as the apical part of the cell and the apical plasma membrane. Regarding molecular functions, the DEEGs were linked to activities such as iron ion, glycosaminoglycan, and sulfur compound binding (Figure 3A–C). KEGG pathway analysis further demonstrated significant enrichment in several critical pathways. These included extracellular matrix (ECM)–receptor interaction, the IL−17 signaling pathway, leukocyte trans-endothelial migration, and the estrogen signaling pathway, which may be crucial in the development of HNSCC (Figure 3D–F).

### 3.3. GSEA Enrichment Analysis

The GSEA results revealed distinct enrichment patterns for the experimental and control groups. Genes in the treatment group were predominantly associated with cell cycle regulation, ECM–receptor interaction, focal adhesion, cancer-related pathways, and small-cell lung cancer (Figure 4A). Conversely, genes of the control group were primary enriched in processes related to drug metabolism via cytochrome P450, histidine metabolism, xenobiotic metabolism by cytochrome P450, ribosome function, valine, leucine, and isoleucine degradation (Figure 4B).

### 3.4. Comparison of the LASSO, SVM-RFE, and Random Forest Algorithms

Univariate logistic regression analysis identified 22 DEEGs as significantly associated with HNSCC risk (*p* < 0.05). Through LASSO regression, 15 DEEGs (*AGRN*, *EXPH5*, *CLIC4*, *TSPAN6*, *TGFBR3*, *FN1*, *MMP9*, *KRT13*, *THY1*, *HBA1*, *FCGBP*, *PFN2*, *BST2*, *EPCAM*, and *PIP*) were identified as critical markers for predicting HNSCC development. LASSO regression path plot visualizes how regression coefficients change as the regularization parameter (lambda) varies (Figure 5A). The coefficients of these genes in the LASSO regression model are illustrated. The cross-validation plot for LASSO regression helps select the optimal lambda value (λ = 15) for regularization (Figure 5B). The SVM algorithm identified ten DEEGs (*KRT13*, *TGFBR3*, *EPCAM*, *PIP*, *HBA1*, *AGRN*, *PFN2*, *TSPAN6*, *MMP9*, and *CEACAM5*) significantly associated with the outcomes. The cross-validation accuracy plot evaluates the relationship between the number of selected features and model accuracy (Figure 5C). Nine DEEGs (*CLIC4*, *AGRN*, *PFN2*, *FN1*, *EXPH5*, *HBA1*, *TSPAN6*, *PIP*, and *MMP9*) were screened by RF analysis (Figure 5D,E). After intersecting the three algorithms, we obtained six key genes (*AGRN*, *TSPAN6*, *MMP9*, *HBA1*, *PFN2*, and *PIP*) with diagnostic value (Figure 5F). The boxplots show that *AGRN*, *TSPAN6*, *MMP9*, *HBA1,* and *PFN2* exhibit significant differences between the control and treatment groups (*p*-value < 0.01) (Figure 5G). The violin plots illustrate the gene expression levels in both the treatment and control groups (Figure 5H–L). *AGRN*, *MMP9*, and *PFN2* show higher expressions in the treatment group (Figure 5H–J), while *TSPAN6* and *HBA1* are expressed at lower levels (Figure 5K,L). In contrast, *PIP* exhibits no noticeable difference between the two groups.

### 3.5. Receiver Operating Characteristic Curve Analysis and a Prognostic Nomogram

ROC curve analysis demonstrated that *AGRN* had the highest predictive accuracy for HNSCC prognosis (AUC = 0.842). It was followed by *PFN2* (AUC = 0.757), *MMP9* (AUC = 0.750), *TSPAN6* (AUC = 0.738), and *HBA1* (AUC = 0.714). In contrast, *PIP* (AUC = 0.482) showed low predictive performance. Therefore, we constructed a diagnostic model for further analysis consisting of *AGRN*, *PFN2*, *MMP9*, *TSPAN6*, and *HBA1* that achieved an impressive AUC of 0.926 (95% CI: 0.884–0.962), showing excellent diagnostic capability (Figure 6A,B).

In order to determine the risk of HNSCC, a prognostic nomogram was created using the expression levels of the identified five essential genes (Figure 6C). Calibration curve analysis confirmed high accuracy of the model in predicting disease outcomes (Figure 6D). DCA further validated the ability of the model to differentiate between control and experimental group samples with high precision. The “Genes” curve falls above both the “All” and “None” curves, indicating that the model provides clinical value within that threshold range (Figure 6E). In the GSE83519 dataset, the diagnostic model exhibited exceptional predictive accuracy. *MMP9* (AUC = 0.961) and *TSPAN6* (AUC = 0.845) showed relatively high predictive performance (Figure 6F,G). The violin plots illustrate gene expression levels in both the treatment and control groups (Figure 6H,I).

### 3.6. ssGSEA and Immune Correlation Analysis

The ssGSEA method was used to examine the distribution of 29 immune cell types in the HNSCC samples. Immune cells, including natural killer cells, regulatory T cells, gamma delta T cells, natural killer T cells, and CD56 bright natural killer cells, were substantially more prevalent in HNSCC tissues than in normal tissues (Figure 7A). Correlation analysis further examined how the expression of six key ERGs related to the immune score. The results showed that *MMP9* had a strong positive correlation with nearly all immune cells, while *AGRN* and *PFN2* were negatively correlated with most immune cells (Figure 7B). These findings imply that they may influence immune responses in HNSCC and play important roles in immune regulation.

### 3.7. Candidate Drug Prediction

Potential therapeutic compounds were identified using the DSigDB database, focusing on genes linked to HNSCC. Based on adjusted *p*-values, the top five candidate drugs were listed: toluidine blue O (TBO), UNII−768N7QO4KH, Hoechst 33258, diphenhydramine, and biochanin A (Figure 8A–E). Notably, *MMP9* and *HBA1* were associated with four of these drugs, while *AGRN* and *MMP9* were linked to biochanin A.

### 3.8. Molecular Docking

The drug binding affinity to their respective target proteins was evaluated using molecular docking. CB-Dock2 software (AutoDock Vina version (1.2.0)) facilitated the prediction of binding sites, interactions, and binding energy. This analysis yielded valid docking results for ten drug–protein complexes (Figure 9A–J). Hydrogen bonds and electrostatic interactions were observed between all drug candidates and their targets, effectively occupying the binding pockets. The compounds C5 and Hoechst 33258 showed the strongest and most stable binding affinity, with the lowest binding energy (−10.2 kcal/mol) among the tested compounds. All binding energies for all drug–target combinations tested in Figure 9A–J are listed in Table 1.

## 4. Discussion

The development of an effective treatment for HNSCC is particularly challenging because the mechanisms of action of new therapeutic approaches are still poorly understood. The intricate structure of the oral–maxillofacial and head and neck regions, which encompasses vital tissues and organs, often renders tissue biopsies both risky and technically challenging. This underscores the pressing need for reliable and specific diagnostic tools, with exosomes emerging as a promising prognostic candidate.

Over the past decade, exosomes have been recognized for their role in transporting diverse bioactive molecules and genetic components that can profoundly influence recipient cell functions and the tumor microenvironment [16]. These characteristics make the bioactivity of exosomes and their cargo valuable potential biomarkers for cancer diagnosis. Researchers need to establish standardized biobanks and analyze patient-specific biomarkers to develop personalized treatment strategies. As potential targets and biomarkers in HNSCC, exosomes hold great promise for advancing personalized medicine.

This study used four GEO datasets to identify disease-specific feature genes in HNSCC, integrating microarray data with batch effect correction. Exosome-related genes retrieved from the GeneCards database were screened for their differential expression, revealing 22 DEGs between high-expression and low-expression groups. Functional and pathway enrichment analyses (GO, KEGG, and GSEA) highlighted the involvement of individual genes in unique biological roles. Machine learning techniques (LASSO, SVM, and RF) identified five key genes (*AGRN*, *TSPAN6*, *MMP9*, *HBA1*, and *PFN2*) that formed the basis of a predictive model, which was validated using ROC and calibration curves. The diagnostic model showed excellent predictive accuracy through external data GSE83519 validation. *MMP9* (AUC = 0.961) and *TSPAN6* (AUC = 0.845) showed relatively high predictive performance. In addition, the ssGSEA algorithm revealed that the three key genes (*MMP9*, *AGRN*, and *PFN2*) share close immunosuppressive, immune response, and key signaling pathways. The analysis of HNSCC gene profiles against the DSigDB database identified potential therapeutic compounds. Accordingly, molecular docking simulations pinpointed ligands with strong binding affinities, with C5 and Hoechst 33258 exhibiting highly stable interactions. These ligands were hence prioritized for further experimental validation and optimization in drug design.

The ECM is a complex network of secreted macromolecules that creates a non-cellular environment, critical for regulating cellular behavior during development, repair, regeneration, tumor progression, and metastasis [17,18,19]. Matrix metalloproteinases (MMPs), part of the metzincin superfamily [20], are key players in ECM remodeling. Among them, MMP−9, also known as gelatinase B, is closely associated with various pathological conditions, including cancer [21]. The dysregulated expression of MMP−9 leads to the degradation of basement membranes and stromal barriers, facilitating tumor invasion. The ECM significantly influences key characteristics of cancer, with changes in its dynamics playing a pivotal role in driving tumor development [22]. In HNSCC, MMP−9 acts as a negative prognostic indicator. It targets various ECM components, breaking down basement membranes and stromal barriers, thereby promoting tumor invasion and spread [23]. The overexpression of MMPs in HNSCC is closely associated with aggressive tumor behavior and the progression of the disease [24,25]. Despite substantial research into the MMP family as prognostic markers and therapeutic targets in HNSCC, the expression levels of MMP members vary across different cell types, highlighting the complexity of their roles in tumor biology. This makes MMP−9 a potential therapeutic target, although no selective inhibitors have successfully advanced through clinical trials.

Agrin (AGRN), a multi-domain protein primarily found in the basement membranes of organs such as the brain, lungs, and muscles [26], can exist as either a membrane-bound or secreted ECM component [27]. In the central nervous system (CNS), AGRN is essential for mediating signaling pathways during neuromuscular synapse formation [28]. It also facilitates acetylcholine receptor clustering, a critical function for neuromuscular junction activity [29]. Despite the limited ability of adult myocardium to regenerate, AGRN has shown potential in stimulating the proliferation and repair of adult cardiomyocytes after a heart attack [30]. Beyond these roles, AGRN has been implicated in numerous human cancers. In hepatocellular carcinoma (HCC), it is expressed and secreted at high levels by tumor cells, promoting their growth and migration. AGRN also acts as a sensor for oncogenic signals during ECM remodeling in hepatic tumors [27]. In oral squamous cell carcinoma (OSCC), elevated agrin levels are observed in both malignant and precancerous tissues [31]. Silencing agrin expression significantly impairs cancer cell behaviors such as movement, growth, invasion, and the formation of colonies and tumor spheroids, highlighting its pathological importance in OSCC progression [31]. However, its exact role in the advancement of HNSCC remains poorly understood and needs further investigation.

Profilin (PFN), a small actin-binding protein of 12–15 kDa, is essential for regulating actin dynamics and is highly conserved across all eukaryotic species [32,33]. PFNs play a key role in diverse cellular functions, including metabolism, signal transduction, cell motility, and gene transcription [34,35]. Profilin 2 (PFN2) has emerged as an essential factor in tumor development and progression within its isoforms [35]. The increased expression of PFN2 expression has been linked to worse prognosis in breast cancer [36] and found to accelerate small-cell lung cancer by promoting tumor-related blood vessel formation [37]. Additionally, PFN2 drives the progression and metastasis of esophageal squamous cell carcinoma (ESCC) [38]. Interestingly, in OSCC, PFN2 exhibits an opposing role by inhibiting tumor growth and aggressiveness [39], highlighting the context-dependent functions of PFN2 in cancer. While its role in various cancers remains underexplored, PFN2 overexpression has been confirmed in HNSCC clinical specimens, where it enhances cancer cell migration and invasion [40]. Furthermore, it facilitates cell growth and metastatic potential in HNSCC by influencing the PI3K/AKT/β-catenin signaling pathway [41]. Despite more research being needed to confirm these observations, PFN2 appears to be a critical regulator in HNSCC progression, offering potential value as both a therapeutic target and disease prognosis biomarker.

Glycosylated hemoglobin (HBA1) forms through a non-enzymatic reaction between glucose and the amino group of hemoglobin. As a widely used indicator for blood glucose testing, HBA1 reflects average blood sugar levels over the past 2–3 months and serves as a critical marker for assessing diabetes control [42]. Elevated HbA1c levels are a significant early warning signal for diabetes and its complications, including peripheral neuropathy, motor dysfunction, sensory loss, cardiovascular disease, and stroke. These complications arise as prolonged high blood sugar accelerates atherosclerosis. Moreover, HbA1c levels have been associated with an increased risk of various cancers, including colorectal, breast, cholangiocarcinoma, and endometrial cancers, particularly in the context of diabetes or metabolic syndrome [43,44,45]. Strikingly, our experimental data suggest that HbA1c could serve as a predictor of both overall and HNSCC risks, regardless of diabetic status. Although the relationship between HbA1 and HNSCC has not been thoroughly investigated, it is evident that diabetes and HNSCC share several common risk factors. However, the exact mechanisms linking these diseases remain unclear. Our findings provide valuable insights that may contribute to the early detection and effective treatment strategies for HNSCC, laying the groundwork for future research.

Tetraspanins (TSPANs), a small transmembrane protein family found in all multicellular organisms, play an important role in a variety of biological processes [46]. These proteins interact with various transmembrane molecules, including adhesion receptors, integrins, cytokine receptors, and growth factors, and modulate cell signaling [47,48]. While the role of TSPAN6 in mammalian systems remains largely unexplored, its mRNA and protein are predominantly expressed in epithelial cells. Studies have shown that reduced TSPAN6 expression is associated with poor survival in lung [48] and pancreatic cancer [49] cohorts exhibiting mesenchymal traits. Additionally, recent findings suggest that TSPAN6 functions as a tumor suppressor in colorectal cancer and may serve as a prognostic biomarker in glioma patients [50]. Based on our results, TSPAN6 could potentially act as a predictive biomarker for HNSCC patients. However, its tumor-suppressive role in HNSCC requires further investigation through future experimental studies.

In summary, AGRN, TSPAN6, MMP9, HBA1, and PFN2 emerge as crucial exosome-carried regulators in HNSCC. These molecules show prognostic significance and are linked to immune modulation, highlighting their potential as biomarkers. While a single biomarker may not provide sufficient diagnostic accuracy, combining multiple biomarkers could enhance precision and capture the complex information conveyed by tumor-derived exosomes. A potential pharmacological approach for targeting AGRN, MMP9, and HBA1 in HNSCC could be developed based on their differential expression between control and treatment cohorts. Inhibiting AGRN through small molecules or neutralizing antibodies may help disrupt its role in tumor invasion and metastasis. Suppressing MMP9 using specific inhibitors, such as small-molecule compounds or monoclonal antibodies, could reduce tumor progression and improve treatment efficacy. Meanwhile, restoring HBA1 expression via gene therapy or epigenetic modulation may help counteract the metabolic and hypoxic advantages of HNSCC cells, potentially limiting tumor survival and progression.

TBO, a basic thiazine metachromatic dye, binds strongly to acidic tissue components such as nucleic acids and polysaccharides, making it useful in histology and clinical diagnostics. Clinically, it aids in detecting oral dysplasia and carcinoma, highlighting structures such as mast cell granules, mucins, and cartilage [51]. TBO is also explored as a photosensitizer in photodynamic therapy (PDT), with studies showing its ability to inhibit methicillin-resistant Staphylococcus aureus biofilms, indicating potential for treating antibiotic-resistant infections [52]. While generally safe in controlled use, TBO may disrupt mitochondrial energy metabolism even without photostimulation, raising safety concerns in PDT [53]. Optimized dosing and application strategies are essential to minimize toxicity.

Hoechst 33258, corresponding to Hoechst dyes, is widely utilized in biological research for DNA staining. However, due to their strong DNA-binding affinity, these dyes can interfere with DNA replication during cell division, potentially leading to mutagenic and carcinogenic effects. Consequently, proper handling and disposal procedures are critical to minimizing associated risks.

Similarly, UNII−768N7QO4KH, corresponding to Trypan blue, is a vital stain commonly employed for cell viability assessment. Research has demonstrated that Trypan blue can exert cytotoxic effects, particularly at elevated concentrations or with prolonged exposure. For example, exposure to concentrations of 0.01% and 0.1% for 24 h has been shown to significantly reduce cell density in human corneal cells [54]. Furthermore, Trypan blue has been implicated in chronic cytotoxicity in human retinal pigment epithelial cells, even at clinically relevant concentrations [55].

Diphenhydramine, a first-generation antihistamine, is widely used for its anticholinergic and sedative effects in treating allergies, insomnia, motion sickness, and Parkinson’s disease symptoms [56]. However, it poses notable risks, including impaired driving performance, sometimes exceeding the effect of alcohol [57], and an increased risk of cognitive decline in older adults [58]. Paradoxical reactions, such as agitation, are reported in children [59], and cases of misuse and dependence have been documented [60]. Given these concerns, its clinical use should be carefully assessed.

Biochanin A, a naturally occurring isoflavone in red clover (Trifolium pratense), chickpeas, soybeans, and other legumes, exhibits anticancer properties [61]. It induces apoptosis, inhibits metastasis, and arrests the cell cycle by targeting dysregulated signaling pathways [62]. Namely, it suppresses prostate cancer cell proliferation without affecting epidermal growth factor receptor tyrosine autophosphorylation [63].

Among the substances listed, diphenhydramine has the most established safety profile in clinical use, with manageable side effects. TBO is also considered safe under controlled conditions. In contrast, Hoechst dyes (UNII−768N7QO4KH and 33258 Hoechst) are limited to research due to cytotoxicity and potential genotoxicity. Biochanin A shows low toxicity in preliminary studies, but a lack of clinical data prevents definitive assessment. Overall, diphenhydramine appears the safest for clinical use, followed by toluidine blue O with proper dosing.

Nonetheless, the clinical application of exosomes faces hurdles, particularly the high costs of isolation and purification. Research must focus on improving these technologies to enable more sensitive, accessible detection methods, facilitating the integration of exosomes into cancer diagnostics and therapy.

## Figures and Tables

**Figure 1 biomedicines-13-00780-f001:**
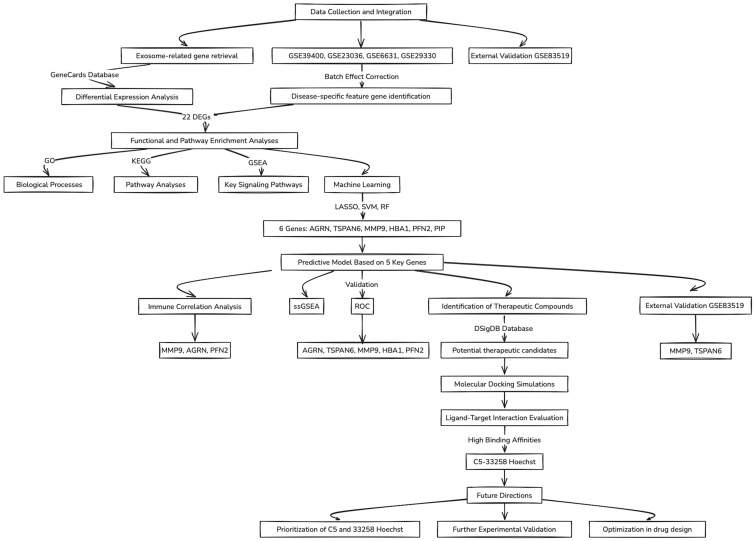
Flowchart of the data analysis workflow.

**Figure 2 biomedicines-13-00780-f002:**
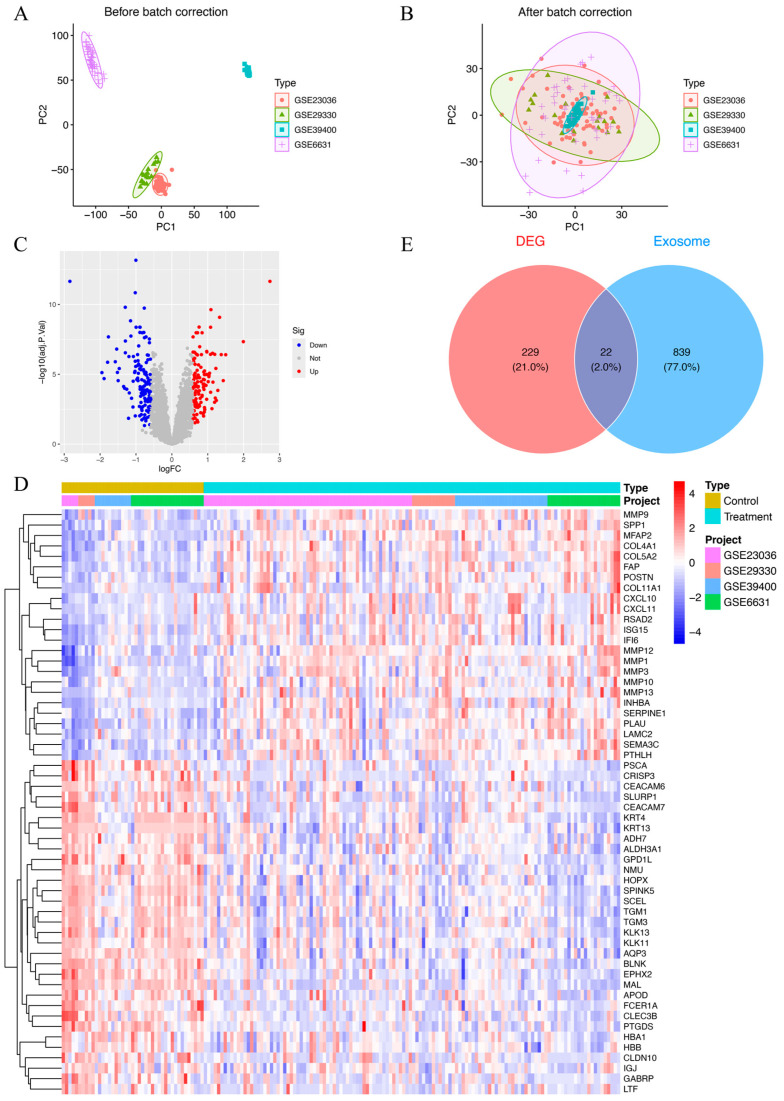
(**A**) Principal component plot of gene expression based on four different GEO datasets before batch correction. (**B**) Principal component plot of gene expression after batch correction on combined datasets (MergeCohort). (**C**) Volcano plot of DEGs between HNSCC patients and healthy controls. In total, 118 upregulated genes and 133 downregulated genes are shown in red and blue, respectively. adj.P.Val, adjusted *p*-value. (**D**) Heatmap of 251 DEGs between HNSCC patients and healthy controls. Red indicates that the gene is upregulated in the treatment group. Blue indicates that the gene is downregulated in the treatment group. The depth of the color reflects the amount of change in expression. (**E**) Venn diagram of 22 DEEGs by intersecting 251 DEEGs and 861 protein-coding ERGs.

**Figure 3 biomedicines-13-00780-f003:**
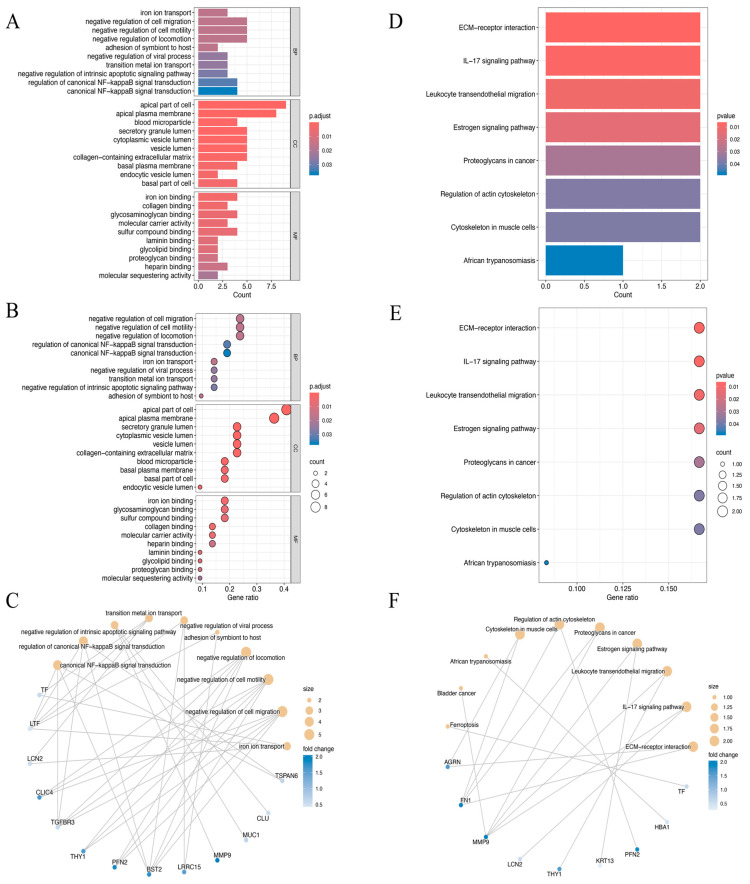
(**A**–**C**) The bar plot (**A**), dot plot (**B**), and cnetplot (**C**) of GO analysis for the top ten biological functional enrichments. p.adjust, *p*-value adjustment. (**D**–**F**) The bar plot (**D**), dot plot (**E**), and cnetplot (**F**) of KEGG analysis for the top ten reactome pathway enrichments.

**Figure 4 biomedicines-13-00780-f004:**
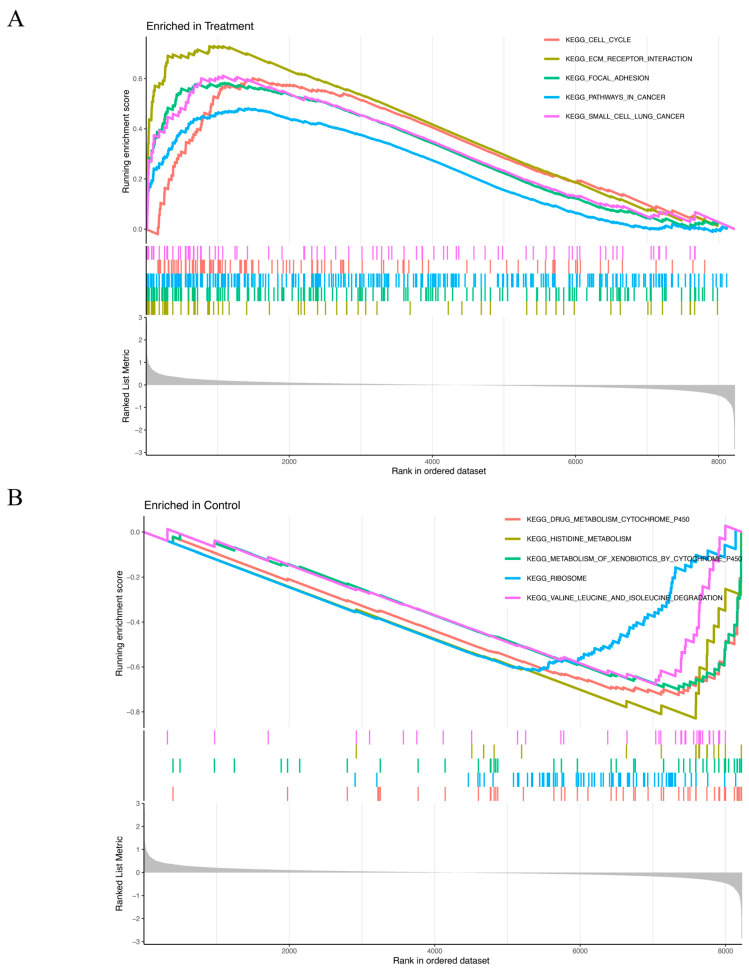
(**A**,**B**) GSEA results show significant enrichment signaling pathways in the treatment group (**A**) and in the control group (**B**). *X*-axis: Rank in ordered dataset represents the position of the gene in the dataset sorted by expression changes. *Y*-axis: Running enrichment score indicates the enrichment of the gene set in the ordered gene list. The curve color represents different KEGG pathways. The colored vertical lines (gene hits) indicate the location of specific genes, which belong to the gene set of the corresponding KEGG pathway. The more concentrated the vertical lines are, the higher the enrichment of genes in the KEGG pathway is in the ranked list. The gray part at the bottom (ranked list metric) reflects the distribution of the ranking scores of genes in the dataset. The closer it is to 0, the less significant the change is. A positive value indicates that the gene is upregulated in the treatment group, and a negative value indicates that the gene is upregulated in the control group. (**A**) Curves of different colors represent different KEGG pathways. The higher the peak of the curve, the more enriched the pathway is in the treatment group. (**B**) Curves of different colors represent different KEGG pathways. The lower the curve, the more enriched the pathway is in the control group.

**Figure 5 biomedicines-13-00780-f005:**
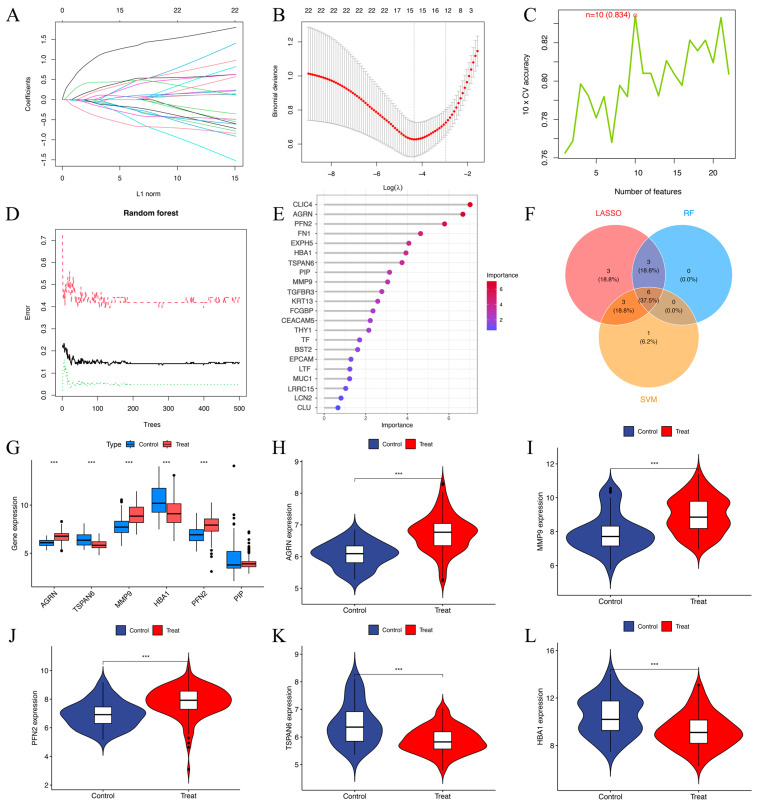
(**A**) The LASSO regression path plot visualizes how regression coefficients change as the regularization parameter (lambda) varies. (**B**) The cross-validation plot for LASSO regression helps select the optimal lambda (λ = 15) value for regularization. (**C**) Cross-validation (CV) accuracy plot indicates that 10 features (n = 10) were selected as optimal. The cross-validation accuracy was 83.4% for this selection. (**D**) shows the error rates for a random forest model as the number of trees in the forest increases. The red dashed line represents the out-of-bag error. The black solid line indicates the training error. The green dotted line represents the test error. (**E**) The screening of 9 key genes using the RF model in 22 DEEGs. The horizontal axis indicates the importance of each gene or protein, while the vertical axis lists different genes or proteins, and their positions in the figure indicate their relative importance. The color change in the bar graph (from blue to red) indicates the importance distribution of each gene. Red indicates a higher importance value, while blue indicates a lower importance value. (**F**) The Venn diagram of 6 target genes generated via the above three algorithms. (**G**–**L**) Boxplots and violin plots of the *AGRN*, *MMP9*, *PFN2*, *TSPAN6,* and *HBA1* gene expression levels between the treatment and control groups. *** *p* < 0.001.

**Figure 6 biomedicines-13-00780-f006:**
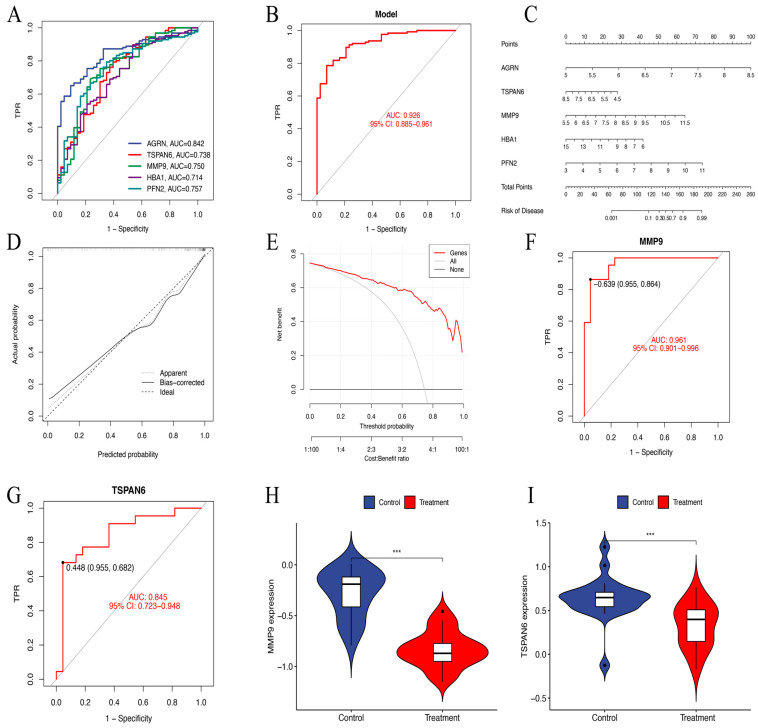
(**A**) The ROC curves of six key genes in LASSO regression analysis. (**B**) The ROC curves of a diagnostic model constructed using the six key genes. AUC = 0.926. TPR, true positive rate. (**C**) Nomogram model predicting HNSCC risk. The nomogram is used by summing all points identified on the scale for each variable. (**D**) The calibration curve for predicting HNSCC risk. The predicted probability by the nomogram model is plotted on the *x*-axis, and the actual probability is plotted on the *y*-axis. (**E**) The “All” curve assumes that all patients are considered high risk and receive intervention. The “None” curve assumes that no intervention is performed on any patient. The “Genes” curve is based on our data-trained predictive model. (**F**,**G**) The ROC nomogram of *MMP9* and *TSPAN6* predicting the prevalence of GSE83519. (**H**,**I**) The violin plots of *MMP9* and *TSPAN6* expression levels between the treatment and control groups in dataset GSE83519. *** *p* < 0.001.

**Figure 7 biomedicines-13-00780-f007:**
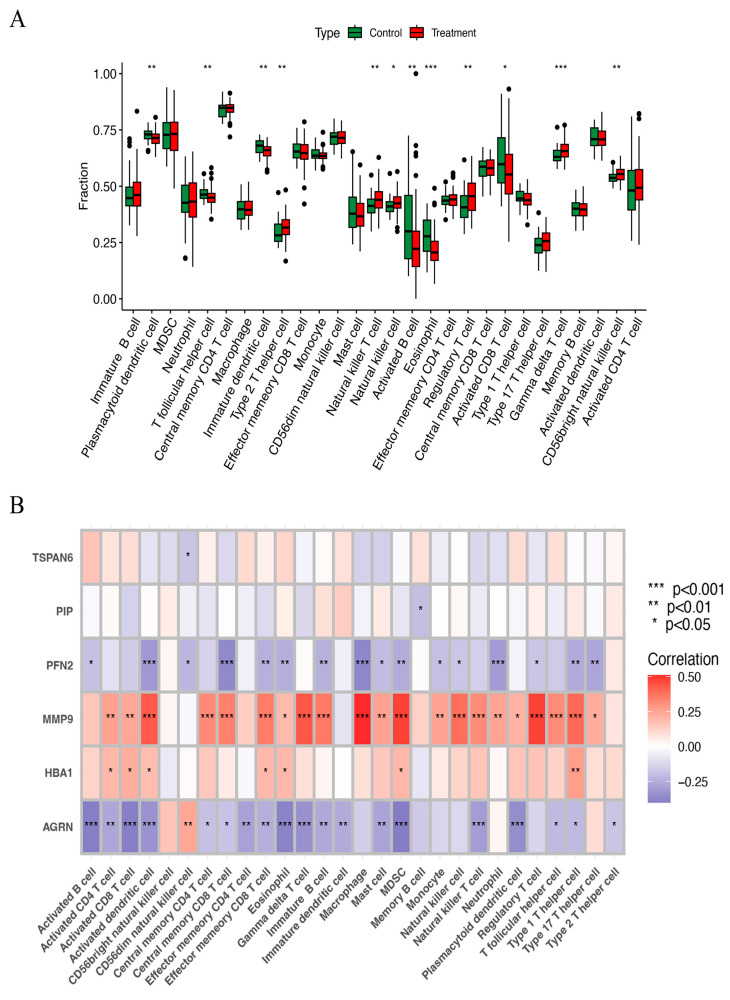
(**A**) The scatter diagram indicates the comparative expression of immune infiltration analysis between the treatment and control groups. * *p* < 0.05, ** *p* < 0.01, and *** *p* < 0.001. (**B**) The bar plot shows the proportions of the 28 different types of immune cells. MDSC, myeloid-derived suppressor cells.

**Figure 8 biomedicines-13-00780-f008:**
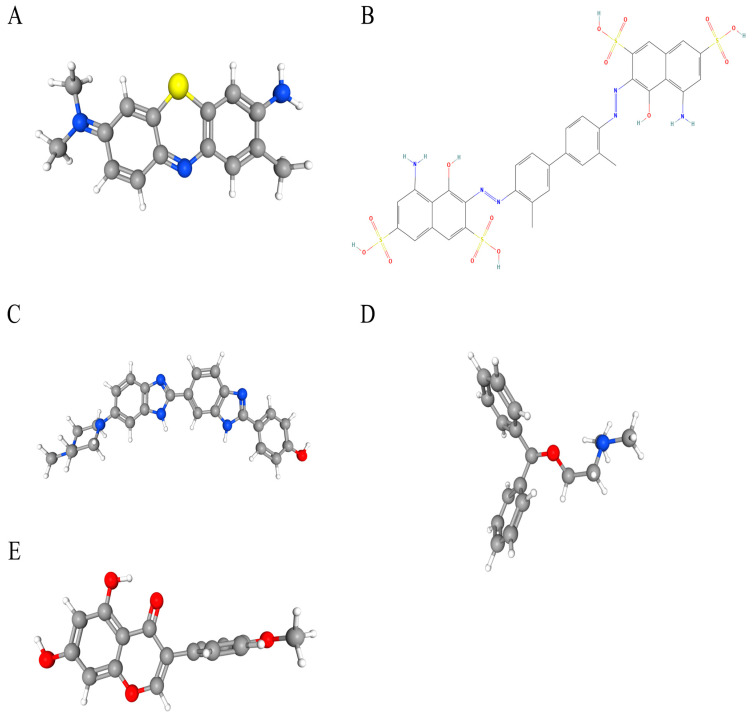
The chemical structures of five candidate drugs. (**A**) toluidine blue O. (**B**) UNII−768N7QO4KH. (**C**) Hoechst 33258. (**D**) diphenhydramine. (**E**) biochanin A.

**Figure 9 biomedicines-13-00780-f009:**
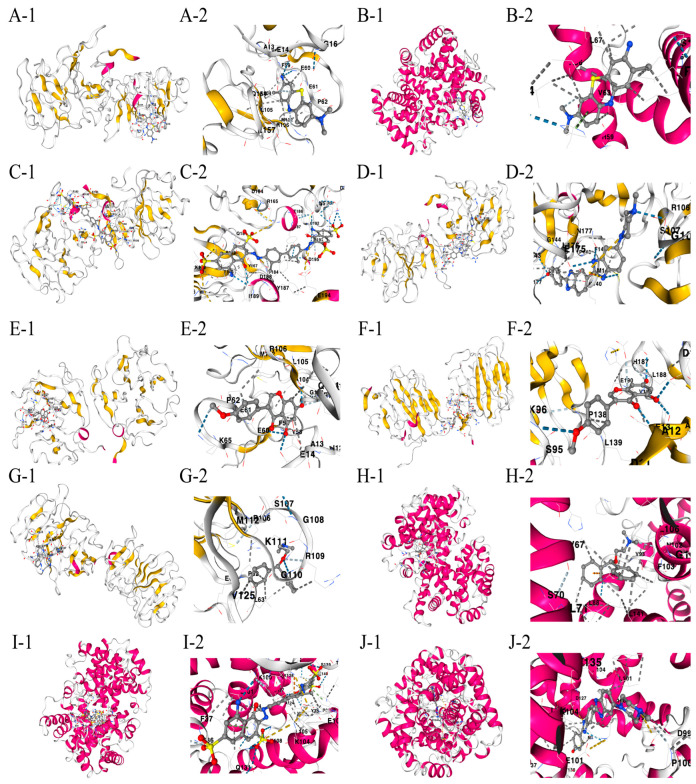
(**A**–**J**) Docking results of available proteins with small molecules. “1” refers to the docking results of the overall view of the complex structure and “2” refers to the detailed zoom narrowing down the precise chemistry of the binding site for each drug-target pair. (**A**) *MMP9* docking toluidine blue O. (**B**) *HBA1* docking toluidine blue O. (**C**) *MMP9* docking UNII−768N7QO4KH. (**D**) *MMP9* docking Hoechst 33258. (**E**) *MMP9* docking biochanin A. (**F**) *AGRN* docking biochanin A. (**G**) *MMP9* docking diphenhydramine. (**H**) *HBA1* docking diphenhydramine. (**I**) *HBA1* docking UNII−768N7QO4KH. (**J**) *HBA1* docking Hoechst 33258.

**Table 1 biomedicines-13-00780-t001:** Binding energies for all drug–target combinations.

Toluidine Blue O—MMP9				
CurPocket ID	Vina Score	Cavity Volume (Å³)	Center (x, y, z)	Docking Size (x, y, z)
C5	−6.8	211	−41, −29, 1	21, 21, 21
C2	−6.2	495	−36, −53, −14	21, 21, 21
C4	−6.2	214	−13, −46, −28	21, 21, 21
C1	−6.1	583	−20, −29, −13	21, 21, 21
C3	−5.7	222	−4, −59, −27	21, 21, 21
Toluidine Blue O—HBA1				
CurPocket ID	Vina score	Cavity volume (Å³)	Center (x, y, z)	Docking size (x, y, z)
C3	−7.9	846	−30, −41, 14	21, 21, 21
C5	−7.9	710	−22, −10, 19	21, 21, 21
C1	−7.7	13258	−22, −26, 5	35, 35, 35
C2	−7.7	870	−7, −20, 5	21, 21, 21
C4	−7.7	787	−38, −27, −2	21, 21, 21
UNII-768N7QO4KH—MMP9				
CurPocket ID	Vina score	Cavity Volume (Å³)	Center (x, y, z)	Docking size (x, y, z)
C2	−8.9	495	−36, −53, −14	35, 35, 35
C1	−8	583	−20, −29, −13	35, 35, 35
C4	−7.7	214	−13, −46, −28	35, 35, 35
C5	−7.7	211	−41, −29, 1	35, 35, 35
C3	−6.7	222	−4, −59, −27	35, 35, 35
Hoechst 33258—MMP9				
CurPocket ID	Vina score	Cavity Volume (Å³)	Center (x, y, z)	Docking size (x, y, z)
C1	−9.7	583	−20, −29, −13	28, 28, 28
C5	−9.4	211	−41, −29, 1	28, 28, 28
C2	−8.7	495	−36, −53, −14	28, 28, 28
C4	−7.4	214	−13, −46, −28	28, 28, 28
C3	−7	222	−4, −59, −27	28, 28, 28
biochanin A—MMP9				
CurPocket ID	Vina score	Cavity Volume (Å³)	Center (x, y, z)	Docking size (x, y, z)
C5	−7.6	211	−41, −29, 1	21, 21, 21
C1	−6.8	583	−20, −29, −13	21, 21, 21
C2	−6.7	495	−36, −53, −14	21, 21, 21
C4	−6	214	−13, −46, −28	21, 21, 21
C3	−5.9	222	−4, −59, −27	21, 21, 21
biochanin A—AGRN				
CurPocket ID	Vina score	Cavity volume (Å³)	Center (x, y, z)	Docking size (x, y, z)
C5	−6.8	128	24, 15, −11	21, 21, 21
C3	−6.3	460	−5, 31, 39	21, 21, 21
C2	−6.1	494	14, 11, 23	21, 21, 21
C4	−5.8	291	19, 17, 20	21, 21, 21
C1	−5.6	670	31, 12, 2	21, 21, 21
diphenhydramine—MMP9				
CurPocket ID	Vina score	Cavity volume (Å³)	Center (x, y, z)	Docking size (x, y, z)
C5	−6.5	211	−41, −29, 1	20, 20, 20
C2	−5.7	495	−36, −53, −14	20, 20, 20
C1	−5.5	583	−20, −29, −13	20, 20, 20
C4	−5.5	214	−13, −46, −28	20, 20, 20
C3	−4.4	222	−4, −59, −27	20, 20, 20
diphenhydramine—HBA1				
CurPocket ID	Vina score	Cavity volume (Å³)	Center (x, y, z)	Docking size (x, y, z)
C5	−8	710	−22, −10, 19	20, 20, 20
C2	−7.5	870	−7, −20, 5	20, 20, 20
C3	−7.5	846	−30, −41, 14	20, 20, 20
C4	−7.4	787	−38, −27, −2	20, 20, 20
C1	−6	13258	−22, −26, 5	35, 35, 35
UNII-768N7QO4KH—HBA1				
CurPocket ID	Vina score	Cavity volume (Å³)	Center (x, y, z)	Docking size (x, y, z)
C1	−10.1	13258	−22, −26, 5	35, 35, 35
C4	−9	787	−38, −27, −2	35, 35, 35
C5	−8.2	710	−22, −10, 19	35, 35, 35
C3	−8.1	846	−30, −41, 14	35, 35, 35
C2	−8	870	−7, −20, 5	35, 35, 35
Hoechst33258—HBA1				
CurPocket ID	Vina score	Cavity volume (Å³)	Center (x, y, z)	Docking size (x, y, z)
C5	−10.2	710	−22, −10, 19	28, 28, 28
C1	−10	13258	−22, −26, 5	35, 35, 35
C2	−10	870	−7, −20, 5	28, 28, 28
C3	−9.8	846	−30, −41, 14	28, 28, 28
C4	−9.6	787	−38, −27, −2	28, 28, 28

## Data Availability

The datasets presented in this study can be found in online repositories. Further inquiries can be directed to the corresponding authors.

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
