# Peer review of "Machine Learning-Driven Identification of Exosome-Related Genes in Head and Neck Squamous Cell Carcinoma for Prognostic Evaluation and Drug Response Prediction"

_biomedicines, 2025, doi:10.3390/biomedicines13040780_

Round 1

Reviewer 1 Report

Comments and Suggestions for Authors

In this manuscript, Cai et. al. identified the HNSCC-related genes through a well-described data preprocessing and machine learning approach. Moreover, the authors also suggested a few drug candidates for these identified proteins using molecular docking. Overall, this is an important work addressing the gap in the field and the manuscript is well-written. I recommend it publishing once the following comments are addressed.  

1. The text in Figure 2D is illegible, and the image resolution appears to be low. Please replace it with a higher-quality figure.

2. In reference to the statement, “The combined datasets, referred to as MergeCohort, exhibited clear separation between HNSCC samples and controls in the PCA plot (Fig. 2B)”: The PCA plot does not appear to show a clear separation, as the data points seem to overlap significantly. Could the authors clarify how this demonstrates a distinct difference?

3.  The term "DEEG" is mentioned without explanation. Could the authors clarify its meaning? Is it a typo for "DEGs"?

4. Could the authors provide a link to the data and code repository for reproducibility?

5. Why is the gene expression level of PIP not shown in Figure 6? Additionally, please specify in the caption what the "***" symbol represents.

Author Response

Dear Editors and Reviewers:

Thank you for your letter and for the reviewers’ comments concerning our manuscript entitled “Machine Learning-Driven Identification of Exosome-Related Genes in Head and Neck Squamous Cell Carcinoma for Prognostic Evaluation and Drug Response Prediction”. Those comments are all valuable and very helpful for revising and improving our paper, as well as important guiding significance to our researches. We have studied comments carefully and have made correction which we hope meet with approval. Revised portion are marked in red in the paper and the respond to the reviewer’s comments are as flowing.

Responds to the reviewer’s comments:

  1. The text in Figure 2D is illegible, and the image resolution appears to be low. Please replace it with a higher-quality figure.

Respond: According to your request, I have adjusted the resolution of Figure 2D to make the text clearer, thanks for your suggestion.

  1. In reference to the statement, “The combined datasets, referred to as MergeCohort, exhibited clear separation between HNSCC samples and controls in the PCA plot (Fig. 2B)”: The PCA plot does not appear to show a clear separation, as the data points seem to overlap significantly. Could the authors clarify how this demonstrates a distinct difference?

Respond: I mistakenly wrote the description for Figure 2A here. The PCA plot in Figure 2B, generated after batch correction analysis, shows that the merged dataset (referred to as MergeCohort) exhibits a uniform distribution between HNSCC samples and controls in the PCA plot rather than a clear separation. I have corrected it, thanks for your suggestion.

  1. The term "DEEG" is mentioned without explanation. Could the authors clarify its meaning? Is it a typo for "DEGs"?

Respond: The term “DEEG” mentioned in line 109 of the text is a misspelling of “DEEGs.” I have corrected it to “Differentially Expressed Exosome-Related Genes”, thanks for your suggestion.

  1. Could the authors provide a link to the data and code repository for reproducibility?

Respond: Thank you for your valuable suggestion regarding reproducibility. We are willing to provide links to the databases used in the article, unfortunately, we are unable to provide a link to code repository at this time due to data sharing restrictions and ongoing proprietary work. However, we would be happy to discuss the methodology and provide additional details upon request to ensure transparency and clarity. The data supporting the findings of this study will subsequently be available from the corresponding author upon reasonable request.

We appreciate your understanding.

  1. Why is the gene expression level of PIP not shown in Figure 6? Additionally, please specify in the caption what the "***" symbol represents.

Respond: In the ROC Curve Analysis validation, the AUC value of PIP was 0.482 (Figure 6A), which is far below the standard threshold of 0.7, indicating its low predictive ability for disease occurrence. Therefore, when constructing the model, five genes with higher predictive capabilities were selected: AGRN, PFN2, MMP9, TSPAN6, and HBA1 (all with AUC values greater than 0.7). The ROC Curve Analysis validation curve of this model is shown in Figure 6B. As a result, Figure 6C does not display the gene expression level of PIP. However, if needed, I can supplement Figure 6C with the gene expression level of PIP.

Additionally, the "***" symbol represents the significance level, p<0.001, we also added the text in line 359, thanks for your suggestion.

We tried best to improve the manuscript and made some changes marked in red in revised paper which will not influence the content and framework of the paper. Once again, thank you very much for your attention and time. Look forward to hearing from you.

Reviewer 2 Report

Comments and Suggestions for Authors

Major points:

1) Very little demographical and clinical information is presented on the control and treatment patient cohorts. Notably, it is not clear what was their average age, how severe was their HNSCC prognosis, what were the common and atypical comorbidities, and how exactly were these patients treated and with what outcome?

2) The genes listed in Figure 2D are impossible to be read. Please increase its size and/or pixel density of this panel.

3) Please either provide all accompanying datasets as part of the supplement or provide link(s) to the online repositories mentioned in "The data sets presented in this study can be found in online repositories" (line 439).

4) The authors have suggested five novel candidate drug modulators for targeting HNSCC by exploiting direct binding interactions with some of the newly discovered exosome-related proteins in Figure 8, however two points necessitate further explanation:

a) Could the authors please briefly comment in the Discussion section on the rationale behind future pharmacological strategy against AGRN, MMP9, HBA1, given the differential expression between the control and treatment patient cohorts these genes display (AGRN and MMP9 upregulated vs HBA1 downregulated) (Figure 5H,I vs 5L, respectively)?

b) Similarly, would the authors please elaborate on the predicted toxicity that "toluidine blue O", "UNII-768N7QO4KH", "33258 Hoechst", "diphenhydramine", and "biochanin A" might be expected to elicit in a clinical setting? Which would be the safest option?

Minor points:

1) Please provide chemical structure illustrations for "toluidine blue O", "UNII-768N7QO4KH", "33258 Hoechst", "diphenhydramine", and "biochanin A" as part of a new figure.

2) Please define abbreviation for "GEO" (line 14), "ROC" (lines 23, 129), "HNSCC" (line 15), "DEEGs" (line 88), "ECM" in "ECM-receptor" (line 220), "SNPs" (line 257).

3) Please format "AGRN" using italics (lines 21, 26, 200, 240, 245, 246, 248, 249, 252, 262, 266, 294, 306, 342, 346).

4) Please format "TSPAN6" using italics (lines 21, 25, 201, 241, 245, 246, 248, 249, 253, 264, 266, 284, 342, 345).

5) Please format "MMP9" using italics (lines 21, 25, 26, 200, 241, 245, 247, 248, 249, 252, 263, 266, 276, 277, 284, 293, 305, 306, 342, 344, 346).

6) Please format "HBA1" using italics (line 21, 202, 241, 245, 246, 248, 249, 253, 264, 266, 305, 342).

7) Please format "PFN2" using italics (lines 21, 26, 201, 241, 245, 246, 248, 249, 252, 263, 266, 294, 342, 346).

8) Please change "33258 Hoechst" to "Hoechst 33258" (lines 31, 304, 313, 319, 321, 350).

9) Please replace "fluids[1].They" with "fluids[1]. They" (line 45).

10) Please change "like" to "such as" (lines 47, 49, 216).

11) The link "https://www.genecards.org" seems to be dysfunctional (line 65). Please fix.

12) Please replace "DEEG Identification" with "Identification of Differentially Expressed Exosome-Related Genes" (line 70).

13) Please replace "https://www.r-project.org/" with "https://www.r-project.org" (line 74).

14) Please change "(version 3.20.0)'s" to "(version 3.20.0)" (line 74).

15) Please change "P-value" to "p-value" (lines 81, 82, 83).

16) Please replace "exosome-related genes" with "ERGs" (lines 88, 100, 197, 209).

17) Please replace "http://bioconductor.org/" with "http://bioconductor.org" (line 92).

18) Please change "ERGs" to "Exosome-Related Genes" (line 105).

19) Please replace "<0.05" with "< 0.05" (line 112).

20) Please change "random-Forest" to "Random Forest" (line 119).

21) Please replace "ROC" with "Receiver Operating Characteristic" (lines 128, 261).

22) Please move "The x-axis displayed the false positive rate (1-specificity), and the y-axis displayed the true positive rate (TPR)" (line 130) into the respective figure legend for plotting ROC.

23) Please change "1-specificity" to "1 - specificity" (line 130).

24) Please replace "trust" with "confidence" (line 132).

25) Please change "ERG" to "Exosome-Related Gene" (line 139).

26) The link "http://dsigdb.tanlab.org/DSigDBv1.0/" seems to be dysfunctional (line 152). Please fix.

27) Please replace "http://dsigdb.tanlab.org/DSigDBv1.0/" with "http://dsigdb.tanlab.org/DSigDBv1.0" (line 152).

28) Please change "http://www.rcsb.org/" to "http://www.rcsb.org" (line 163).

29) Please replace "Analysis" with "Analyses" (line 171).

30) Please change "was" to "is" (line 180).

31) Please replace "DEEGs" with "Differentially Expressed Exosome-Related Genes" (lines 184, 210).

32) Please replace "post-normalization and" with "post-normalization" (line 191).

33) Please change "differences" to differences (Fig. 2B)" (line 193).

34) Please replace "2C-D" with "2C, D" (line 197).

35) Please format "CLIC4" using italics (lines 200, 241, 246).

36) Please format "FN1" using italics (lines 200, 241, 246).

37) Please format "THY1" using italics (line 200, 241).

38) Please format "BST2" using italics (lines 201, 241).

39) Please format "EPCAM" using italics (lines 201, 241, 244).

40) Please format "LRRC15" using italics (line 201).

41) Please format "MUC1" using italics (line 201).

42) Please format "EXPH5" using italics (lines 202, 240, 246).

43) Please format "TF" using italics (line 202).

44) Please format "PIP" using italics (lines 202, 242, 244, 246, 248, 254, 264).

45) Please format "LCN2" using italics (line 202).

46) Please format "TGFBR3" using italics (lines 202, 241, 244).

47) Please format "CEACAM5" using italics (lines 202, 245).

48) Please format "FCGBP" using italics (lines 202, 241).

49) Please format "CLU" using italics (line 202).

50) Please format "LTF" using italics (line 202).

51) Please format "KRT13" using italics (lines 202, 241, 244).

52) "A diagram of a diagram of a diagram Description automatically generated with medium confidence" sign appears when hovering the mouse cursor over Figure 2. Please disable this feature.

53) Please change "Diagram" to "diagram" (line 208).

54) Please replace "3A-C" with "3A–C" (line 219).

55) Please change "3D-F" to "3D–F" (line 222).

56) "A close-up of a graph Description automatically generated" sign appears when hovering the mouse cursor over Figure 3. Please disable this feature.

57) Please change "A-C" to "A–C" (line 224).

58) Please replace "D-F" with "D–F" (line 225).

59) "A close-up of a graph Description automatically generated" sign appears when hovering the mouse cursor over Figure 4. Please disable this feature.

60) Please change "Enriched in Treat" to "Enriched in Treatment" in Figure 4A.

61) Please replace "Running Enrichment Score" with "Running enrichment score" and "Rank in Ordered Dataset" with "Rank in ordered dataset" in Figure 4A, B.

62) Please describe the meaning of Figure 4 in its respective legend. What is "Running Enrichment Score", "Ranked List Metric", "Rank in Ordered Dataset"? What do the colored bars underneath the "Running Enrichment Score" traces indicate?

63) Please change "treat" to "treatment" (lines 236, 250).

64) The authors claim that "Through LASSO regression, 15 DEEGs (AGRN, EXPH5, CLIC4, TSPAN6, TGFBR3, FN1, MMP9, KRT13, THY1, HBA1, FCGBP, PFN2, BST2, EPCAM, and PIP) were identified as critical markers for predicting HNSCC development (Fig. 5A)" (line 240), however none of these genes seems to be shown in Figure 5A.

65) The authors argue that "The coefficients of these genes in the LASSO regression model are illustrated in Figure (Fig. 5B)" (line 243), however there seem to be no coefficients depicted in Figure 5B.

66) Please replace "illustrated in Figure" with "illustrated" (line 243).

67) The authors state that "The SVM algorithm identified 10 DEEGs (KRT13, TGFBR3, EPCAM, PIP, HBA1, AGRN, PFN2, TSPAN6, MMP9, and CEACAM5) significantly associated with the outcomes (Fig. 5C)" (line 244), despite the fact that no DEEGs are shown in Figure 5C. This is rather puzzling as no description seems to be provided.

68) The authors claim that "9 DEEGs (CLIC4, AGRN, PFN2, FN1, EXPH5, HBA1, TSPAN6, PIP, and MMP9) were screened by RF analysis (Fig. 5D-E)" (line 246), but only "TSPAN5" and not "TSPAN6" seems to be shown in Figure 5E. Please resolve this dichotomy.

69) Please change "5D-E" to "5D, E" (line 247).

70) Please replace "MMP9, PFN2, TSPAN6, and HBA1" with "TSPAN6, MMP9, HBA1, and PFN2" (line 249).

71) Please change "groups" to "groups (Fig. 5H–L)" (line 252).

72) Please replace "5H-J" with "5H–J" (line 253).

73) Please change "5K-L" to "5K, L" (line 253).

74) "A collage of graphs and diagrams Description automatically generated" sign appears when hovering the mouse cursor over Figure 5. Please disable this feature.

75) Figure 5A is puzzling as it seems to contain two x-axes. How can one associate the plotted traces to the respective x-axes?

76) Please describe Figure 5A in greater detail in its figure legend. What are the "Coefficients" and what is "L1 Norm"? What does "Norm" stand for?

77) Please replace "L1 Norm" with "L1 norm" in the x-axis title of Figure 5A.

78) Figure 5B seems to also contain two different x-axes. How can one distinguish which data point belongs to which x-axis?

79) Please change "Binomial Deviance" to "Binomial deviance" in the y-axis title of Figure 5B.

80) From Figure 5C is not easily evident what does "10 x CV Accuracy" and "Number of Features" refer to? Moreover, the significance of "0.834" is not clear. Please explain in the figure legend.

81) Please replace "10 x CV Accuracy" with "10 x CV accuracy" and "Number of Features" with "Number of features" in the y- and x-axis title of Figure 5C, respectively.

82) Please provide complete description of Figure 5D in the respective figure legend as it is not clear what do the red and green dashed lines plus the black continuous line indicate? Moreover, what do "Error" and "trees" reflect?

83) Please replace "trees" with "Trees" in the x-axis title of Figure 5D.

84) From the legend to Figure 5B is not clear what is "Binomial Deviance" and "λ"? Please indicate also what the two vertical gray dashed lines highlight?

85) From the legend to Figure 5E is not clear what does "NA" stand for?

86) The authors claim that "15 variables were selected after 10-fold cross-validation of the results" (line 256), however this is not directly obvious from Figure 5A. Where is the "10-fold cross-validation of the results" shown? Which are the 15 variables that "were selected"? Please explain in the figure legend itself.

87) The authors state that "LASSO coefficient profiles of 15 significant SNPs" (line 256) without any obvious "LASSO coefficient profiles" shown in the underlying Figure 5B. Also, what are the "15 significant SNPs"? Please make this figure more comprehensible by providing context in the respective figure legend.

88) Although the authors argue that "Screening of 10 key genes using the SVM model in 22 DEEGs" (line 257), no genes nor/ DEEGs are displayed in the underpinning Figure 5C. This limits the ability of general readers to understand.

89) Please change "genes" to something like "genes generated" or "genes obtained" (line 259).

90) Please replace "6A-B" with "6A, B" (line 268).

91) "A collage of diagrams Description automatically generated" sign appears when hovering the mouse cursor over Figure 6. Please disable this feature.

92) Figure 6E is rather puzzling as the definition difference between "Genes", "All", and "None" seems to be missing. Please explain in the respective figure legend.

93) Please change "Net Benefit" to "Net benefit" and "Cost:Benefit Ratio" to "Cost:Benefit ratio" in the y- and x-axis title of Figure 6E, respectively.

94) Please replace "Treat" with "Treatment" in Figure 6H, I 2x.

95) From the legend to Figure 6D is not clear what is the difference between "Apparent", "Bias-corrected", and "Ideal" calibration curve?

96) From the legend to Figure 6E is not clear what is "Net Benefit" and "Cost:Benefit Ratio"?

97) Please explain the relationship between "Threshold probability" and the "Cost:Benefit Ratio" in the legend to Figure 6E. Threshold for what?

98) Please change "6" to "six" (line 270).

99) Please replace "using" with "using the" (line 271).

100) The authors claim "AUC = 0.809" (line 271), however this value is plotted as 0.926 in Figure 6B. Please fix.

101) Please change "TSPAN" to "TSPAN6" (lines 276, 277).

102) Please format "TSPAN" using italics (lines 276, 277).

103) Please change "genes" to "genes (Figure 6C)" (line 280).

104) Please replace "outcomes" with "outcomes (Figure 6D)" (line 281).

105) Please change "Decision curve analysis (DCA)" to "DCA" (line 281).

106) Please replace "6C-E" with "6E" (line 283).

107) Please change "6F-G" to "6F, G" (line 285).

108) Please replace "illustrate the" with "illustrate" (line 285).

109) Please change "6H-I" to "6H, I" (line 286).

110) "A close-up of a graph Description automatically generated" sign appears when hovering the mouse cursor over Figure 7. Please disable this feature.

111) Please replace "Treat" with "Treatment" in Figure 7A.

112) Please define abbreviation for "MDSC" in the legend to Figure 7.

113) The authors posit that "The bar plot shows the proportions of the 29 different types of immune cells" (line 299), despite the fact that 28 immune cell types are shown in Figure 7B. Please correct the count.

114) Please change "8A-J"t to "8A–J" (line 311).

115) Please provide reference for "Over the past decade, exosomes have been recognized for their role in transporting diverse bioactive molecules and genetic components that can profoundly influence recipient cell functions and the tumor microenvironment" (line 329).

116) Please change "GSEA" to "and GSEA" (line 340).

117) Please replace "RF" with "and RF" (line 341).

118) Please change "The overexpression" to "Overexpression" (line 363).

119) Please provide reference for "In oral squamous cell carcinoma (OSCC), elevated agrin levels are observed in both malignant and precancerous tissues" (line 379).

120) Please replace "PI3K/AKT/β-Catenin" with "the PI3K/AKT/β-catenin" (line 398).

Author Response

Dear Editors and Reviewers:

Thank you for your letter and for the reviewers’ comments concerning our manuscript entitled “Machine Learning-Driven Identification of Exosome-Related Genes in Head and Neck Squamous Cell Carcinoma for Prognostic Evaluation and Drug Response Prediction”. Those comments are all valuable and very helpful for revising and improving our paper, as well as important guiding significance to our researches. We have studied comments carefully and have made correction which we hope meet with approval. Revised portion are marked in red in the paper and the respond to the reviewer’s comments are as flowing.

Responds to the reviewer’s comments:

Major points:

1) Very little demographical and clinical information is presented on the control and treatment patient cohorts. Notably, it is not clear what was their average age, how severe was their HNSCC prognosis, what were the common and atypical comorbidities, and how exactly were these patients treated and with what outcome?
Respond: We extracted the clinical data information of the control and treatment patient cohorts from the GEO datasets GSE39400, GSE23036, GSE6631, GSE29330, and GSE83519 and provided them to review.

2) The genes listed in Figure 2D are impossible to be read. Please increase its size and/or pixel density of this panel.
Respond: We have increased its size in new Figure 2D, thanks for your suggestion.

3) Please either provide all accompanying datasets as part of the supplement or provide link(s) to the online repositories mentioned in "The data sets presented in this study can be found in online repositories" (line 439).
Respond: Thank you for your valuable suggestion regarding reproducibility. We are willing to provide links to the databases used in the article, unfortunately, we are unable to provide a link to code repository at this time due to data sharing restrictions and ongoing proprietary work. However, we would be happy to discuss the methodology and provide additional details upon request to ensure transparency and clarity. The data supporting the findings of this study will subsequently be available from the corresponding author upon reasonable request.

We appreciate your understanding.

4) The authors have suggested five novel candidate drug modulators for targeting HNSCC by exploiting direct binding interactions with some of the newly discovered exosome-related proteins in Figure 8, however two points necessitate further explanation:

a) Could the authors please briefly comment in the Discussion section on the rationale behind future pharmacological strategy against AGRN, MMP9, HBA1, given the differential expression between the control and treatment patient cohorts these genes display (AGRN and MMP9 upregulated vs HBA1 downregulated) (Figure 5H,I vs 5L, respectively)?
Respond: We have briefly explained future pharmacological strategy against AGRN, MMP9 and HBA1 respectively in the Discussion section (lines 545–553).

  1. b) Similarly, would the authors please elaborate on the predicted toxicity that "toluidine blue O", "UNII-768N7QO4KH", "33258 Hoechst", "diphenhydramine", and "biochanin A" might be expected to elicit in a clinical setting? Which would be the safest option?
    Respond: We elaborated on the predicted toxicities that “Toluidine Blue O”, “UNII-768N7QO4KH”, “33258 Hoechst”, “Diphenhydramine”, and “Bichonin A” may cause in the clinical setting and provided medication recommendations based on drug safety (lines 554–587).

Minor points:

1) Please provide chemical structure illustrations for "toluidine blue O", "UNII-768N7QO4KH", "33258 Hoechst", "diphenhydramine", and "biochanin A" as part of a new figure.
Respond: We made a new figure (Figure 8) to illustrate chemical structure for "toluidine blue O", "UNII-768N7QO4KH", "33258 Hoechst", "diphenhydramine", and "biochanin A", so the original Figure8 is changed to Figure9.

2) Please define abbreviation for "GEO" (line 14), "ROC" (lines 23, 129), "HNSCC" (line 15), "DEEGs" (line 88), "ECM" in "ECM-receptor" (line 220), "SNPs" (line 257).
Respond: We have defined abbreviation for "GEO" (line 14), "ROC" (lines 22, 149), "HNSCC" (line 15), "DEEGs" (line 108), "ECM" in "ECM-receptor" (line 268), the "SNPs" (line 257) was deleted, and we have modified the figure legend, thanks for your suggestion.

3) Please format "AGRN" using italics (lines 21, 26, 200, 240, 245, 246, 248, 249, 252, 262, 266, 294, 306, 342, 346).
Respond: We have formatted "AGRN" using italics (lines 26, 29, 240, 293, 300, 303, 305, 306, 309, 336, 347, 384, 401, 422, 445, 449), thanks for your suggestion.

4) Please format "TSPAN6" using italics (lines 21, 25, 201, 241, 245, 246, 248, 249, 253, 264, 266, 284, 342, 345).
Respond: We have formatted "TSPAN6" using italics (lines 26, 27, 241, 294, 300, 303, 305, 306, 310, 345, 347, 358, 359, 374, 445, 448), thanks for your suggestion.

5) Please format "MMP9" using italics (lines 21, 25, 26, 200, 241, 245, 247, 248, 249, 252, 263, 266, 276, 277, 284, 293, 305, 306, 342, 344, 346).
Respond: We have formatted "MMP9" using italics (lines 26, 27, 28, 240, 294, 300, 303, 305, 306, 309, 337, 347, 358, 359, 374, 383, 400, 401, 420, 421, 422, 445, 448, 449), thanks for your suggestion.

6) Please format "HBA1" using italics (line 21, 202, 241, 245, 246, 248, 249, 253, 264, 266, 305, 342).

Respond: We have formatted "HBA1" using italics (lines 26, 242, 294, 300, 303, 305, 306, 310, 345, 347, 400, 445), thanks for your suggestion.

7) Please format "PFN2" using italics (lines 21, 26, 201, 241, 245, 246, 248, 249, 252, 263, 266, 294, 342, 346).
Respond: We have formatted "PFN2" using italics (lines 26, 29, 240, 294, 300, 303, 305, 306, 309, 337, 347, 384, 445, 450), thanks for your suggestion.

8) Please change "33258 Hoechst" to "Hoechst 33258" (lines 31, 304, 313, 319, 321, 350).
Respond: We have changed "33258 Hoechst" to "Hoechst 33258" (lines 33, 399, 403, 411, 422, 424, 453), thanks for your suggestion.

9) Please replace "fluids[1].They" with "fluids[1]. They" (line 45).
Respond: We have replaced "fluids[1].They" with "fluids[1]. They" (line 56), thanks for your suggestion.

10) Please change "like" to "such as" (lines 47, 49, 216).
Respond: We have changed "like" to "such as" (lines 58, 60, 256), thanks for your suggestion.

11) The link "https://www.genecards.org" seems to be dysfunctional (line 65). Please fix.
Respond: We have modified the link and it can now open normally (line 76).

12) Please replace "DEEG Identification" with "Identification of Differentially Expressed Exosome-Related Genes" (line 70).
Respond: We have replaced "DEEG Identification" with "Identification of Differentially Expressed Exosome-Related Genes" (line 81), thanks for your suggestion.

13) Please replace "https://www.r-project.org/" with "https://www.r-project.org" (line 74).
Respond: We have replaced "https://www.r-project.org/" with "https://www.r-project.org" (line 85), thanks for your suggestion.

14) Please change "(version 3.20.0)'s" to "(version 3.20.0)" (line 74).
Respond: We have changed "(version 3.20.0)'s" to "(version 3.20.0)" (line 85), thanks for your suggestion.

15) Please change "P-value" to "p-value" (lines 81, 82, 83).
Respond: We have changed "P-value" to "p-value" (lines 92, 93, 94), thanks for your suggestion.

16) Please replace "exosome-related genes" with "ERGs" (lines 88, 100, 197, 209).
Respond: We have replaced "exosome-related genes" with "ERGs" (lines 108, 120, 231, 249), thanks for your suggestion.

17) Please replace "http://bioconductor.org/" with "http://bioconductor.org" (line 92).
Respond: We have replaced "http://bioconductor.org/" with "http://bioconductor.org" (line 112), thanks for your suggestion.

18) Please change "ERGs" to "Exosome-Related Genes" (line 105).
Respond: We have changed "ERGs" to "Exosome-Related Genes" (line 125), thanks for your suggestion.

19) Please replace "<0.05" with "< 0.05" (line 112).
Respond: We have replaced "<0.05" with "< 0.05" (line 132), thanks for your suggestion.

20) Please change "random-Forest" to "Random Forest" (line 119).
Respond: We have changed "random-Forest" to "Random Forest" (line 139), thanks for your suggestion.

21) Please replace "ROC" with "Receiver Operating Characteristic" (lines 128, 261).
Respond: We have replaced "ROC" with "Receiver Operating Characteristic" (lines 148, 335), thanks for your suggestion.

22) Please move "The x-axis displayed the false positive rate (1-specificity), and the y-axis displayed the true positive rate (TPR)" (line 130) into the respective figure legend for plotting ROC.
Respond: We have moved "The x-axis displayed the false positive rate (1-specificity), and the y-axis displayed the true positive rate (TPR)" (line 151) into the respective figure legend for plotting ROC, thanks for your suggestion.

23) Please change "1-specificity" to "1 - specificity" (line 130).
Respond: We have changed "1-specificity" to "1 - specificity" (line 151), thanks for your suggestion.

24) Please replace "trust" with "confidence" (line 132).
Respond: We have replaced "trust" with "confidence" (line 161), thanks for your suggestion.

25) Please change "ERG" to "Exosome-Related Gene" (line 139).
Respond: We have changed "ERG" to "Exosome-Related Gene" (line 168), thanks for your suggestion.

26) The link "http://dsigdb.tanlab.org/DSigDBv1.0/" seems to be dysfunctional (line 152). Please fix.
Respond: We have modified the link and it can now open normally (line 182).

27) Please replace "http://dsigdb.tanlab.org/DSigDBv1.0/" with "http://dsigdb.tanlab.org/DSigDBv1.0" (line 152).
Respond: We have replaced "http://dsigdb.tanlab.org/DSigDBv1.0/" with "http://dsigdb.tanlab.org/DSigDBv1.0" (line 182), thanks for your suggestion.

28) Please change "http://www.rcsb.org/" to "http://www.rcsb.org" (line 163). 
Respond: We have changed "http://www.rcsb.org/" to "http://www.rcsb.org" (line 191), thanks for your suggestion.

29) Please replace "Analysis" with "Analyses" (line 171).
Respond: We have replaced "Analysis" with "Analyses" (line 200), thanks for your suggestion.

30) Please change "was" to "is" (line 180).
Respond: We have changed "was" to "is" (line 214), thanks for your suggestion.

31) Please replace "DEEGs" with "Differentially Expressed Exosome-Related Genes" (lines 184, 210).
Respond: We have replaced "DEEGs" with "Differentially Expressed Exosome-Related Genes" (lines 218, 250), thanks for your suggestion.

32) Please replace "post-normalization and" with "post-normalization" (line 191).
Respond: We have replaced "post-normalization and" with "post-normalization" (line 225), thanks for your suggestion.

33) Please change "differences" to differences (Fig. 2B)" (line 193).
Respond: We have changed "differences" to differences (Fig. 2B)" (line 227), thanks for your suggestion.

34) Please replace "2C-D" with "2C, D" (line 197).
Respond: We have replaced "2C-D" with "2C, D" (line 231), thanks for your suggestion.

35) Please format "CLIC4" using italics (lines 200, 241, 246).
Respond: We have formatted "CLIC4" using italics (lines 240, 293, 303), thanks for your suggestion.

36) Please format "FN1" using italics (lines 200, 241, 246).
Respond: We have formatted "FN1" using italics (lines 240, 294, 303), thanks for your suggestion.

37) Please format "THY1" using italics (line 200, 241).
Respond: We have formatted "THY1" using italics (line 240, 294), thanks for your suggestion.

38) Please format "BST2" using italics (lines 201, 241).
Respond: We have formatted "BST2" using italics (line 240, 294), thanks for your suggestion.

39) Please format "EPCAM" using italics (lines 201, 241, 244).
Respond: We have formatted "EPCAM" using italics (lines 240, 294, 300), thanks for your suggestion.

40) Please format "LRRC15" using italics (line 201).
Respond: We have formatted "LRRC15" using italics (line 241), thanks for your suggestion.

41) Please format "MUC1" using italics (line 201).
Respond: We have formatted "MUC1" using italics (line 241), thanks for your suggestion.

42) Please format "EXPH5" using italics (lines 202, 240, 246).
Respond: We have formatted "EXPH5" using italics (lines 241, 293, 303), thanks for your suggestion.

43) Please format "TF" using italics (line 202).
Respond: We have formatted "TF" using italics (line 241), thanks for your suggestion.

44) Please format "PIP" using italics (lines 202, 242, 244, 246, 248, 254, 264).
Respond: We have formatted "PIP" using italics (lines 241, 295, 300, 303, 305, 310, 345), thanks for your suggestion.

45) Please format "LCN2" using italics (line 202).
Respond: We have formatted "LCN2" using italics (line 242), thanks for your suggestion.

46) Please format "TGFBR3" using italics (lines 202, 241, 244).
Respond: We have formatted "TGFBR3" using italics (lines 242, 294, 300), thanks for your suggestion.

47) Please format "CEACAM5" using italics (lines 202, 245).
Respond: We have formatted "CEACAM5" using italics (lines 242, 300), thanks for your suggestion.

48) Please format "FCGBP" using italics (lines 202, 241).
Respond: We have formatted "FCGBP" using italics (lines 242, 294), thanks for your suggestion.

49) Please format "CLU" using italics (line 202).
Respond: We have formatted "CLU" using italics (lines 242), thanks for your suggestion.

50) Please format "LTF" using italics (line 202).
Respond: We have formatted "LTF" using italics (lines 242), thanks for your suggestion.

51) Please format "KRT13" using italics (lines 202, 241, 244).
Respond: We have formatted "KRT13" using italics (lines 242, 294, 300), thanks for your suggestion.

52) "A diagram of a diagram of a diagram Description automatically generated with medium confidence" sign appears when hovering the mouse cursor over Figure 2. Please disable this feature.
Respond: We have disabled this feature, thanks for your suggestion.

53) Please change "Diagram" to "diagram" (line 208).
Respond: We have changed "Diagram" to "diagram" (line 248), thanks for your suggestion.

54) Please replace "3A-C" with "3A–C" (line 219).
Respond: We have replaced "3A-C" with "3A–C" (line 267), thanks for your suggestion.

55) Please change "3D-F" to "3D–F" (line 222).
Respond: We have changed "3D-F" to "3D–F" (line 271), thanks for your suggestion.

56) "A close-up of a graph Description automatically generated" sign appears when hovering the mouse cursor over Figure 3. Please disable this feature.
Respond: We have disabled this feature, thanks for your suggestion.

57) Please change "A-C" to "A–C" (line 224).
Respond: We have changed "A-C" to "A–C" (line 273), thanks for your suggestion.

58) Please replace "D-F" with "D–F" (line 225).
Respond: We have replaced "D-F" with "D–F" (line 274), thanks for your suggestion.

59) "A close-up of a graph Description automatically generated" sign appears when hovering the mouse cursor over Figure 4. Please disable this feature.
Respond: We have disabled this feature, thanks for your suggestion.

60) Please change "Enriched in Treat" to "Enriched in Treatment" in Figure 4A.
Respond: We have changed "Enriched in Treat" to "Enriched in Treatment" in Figure 4A.

61) Please replace "Running Enrichment Score" with "Running enrichment score" and "Rank in Ordered Dataset" with "Rank in ordered dataset" in Figure 4A, B.
Respond: We have replaced "Running Enrichment Score" with "Running enrichment score" and "Rank in Ordered Dataset" with "Rank in ordered dataset" in Figure 4A, B.

62) Please describe the meaning of Figure 4 in its respective legend. What is "Running Enrichment Score", "Ranked List Metric", "Rank in Ordered Dataset"? What do the colored bars underneath the "Running Enrichment Score" traces indicate?
Respond: The Running Enrichment Score tracks the accumulation of genes from a set along a ranked list.The Ranked List Metric determines the ranking order of genes based on their correlation with a phenotype.The Rank in Ordered Dataset represents each gene’s position in this ranked list.The Colored Bars show where genes from the gene set appear in the ranking, helping to visualize enrichment patterns.

63) Please change "treat" to "treatment" (lines 236, 250).
Respond: We have changed "treat" to "treatment" (lines 289, 307), thanks for your suggestion.

64) The authors claim that "Through LASSO regression, 15 DEEGs (AGRN, EXPH5, CLIC4, TSPAN6, TGFBR3, FN1, MMP9, KRT13, THY1, HBA1, FCGBP, PFN2, BST2, EPCAM, and PIP) were identified as critical markers for predicting HNSCC development (Fig. 5A)" (line 240), however none of these genes seems to be shown in Figure 5A.
Respond: I have modified the description of the Figure 5A.

65) The authors argue that "The coefficients of these genes in the LASSO regression model are illustrated in Figure (Fig. 5B)" (line 243), however there seem to be no coefficients depicted in Figure 5B.
Respond: Figure 5B is a cross-validation plot for LASSO regression, generated using the package “glmnet” in R. This plot helps select the optimal lambda (λ) value for regularization. In Figure 5B, the best λ is 15. I have modified the annotation of Figure 5B.

66) Please replace "illustrated in Figure" with "illustrated" (line 243).
Respond: We have replaced "illustrated in Figure" with "illustrated" (line 298), thanks for your suggestion.

67) The authors state that "The SVM algorithm identified 10 DEEGs (KRT13, TGFBR3, EPCAM, PIP, HBA1, AGRN, PFN2, TSPAN6, MMP9, and CEACAM5) significantly associated with the outcomes (Fig. 5C)" (line 244), despite the fact that no DEEGs are shown in Figure 5C. This is rather puzzling as no description seems to be provided.
Respond: Figure 5C indicates that ten features (n=10) were selected as optimal which means using ten features provides the best balance between model performance and feature complexity. We have modified the description (line 330–332).

68) The authors claim that "9 DEEGs (CLIC4, AGRN, PFN2, FN1, EXPH5, HBA1, TSPAN6, PIP, and MMP9) were screened by RF analysis (Fig. 5D-E)" (line 246), but only "TSPAN5" and not "TSPAN6" seems to be shown in Figure 5E. Please resolve this dichotomy.
Respond: We checked the figure, and the gene shown in Figure 5E was identified as TSPAN6.

69) Please change "5D-E" to "5D, E" (line 247).
Respond: We have changed "5D-E" to "5D, E" (line 304), thanks for your suggestion.

70) Please replace "MMP9, PFN2, TSPAN6, and HBA1" with "TSPAN6, MMP9, HBA1, and PFN2" (line 249).
Respond: We have replaced "MMP9, PFN2, TSPAN6, and HBA1" with "TSPAN6, MMP9, HBA1, and PFN2" (line 306), thanks for your suggestion.

71) Please change "groups" to "groups (Fig. 5H–L)" (line 252).
Respond: We have changed "groups" to "groups (Fig. 5H–L)" (line 309), thanks for your suggestion.

72) Please replace "5H-J" with "5H–J" (line 253).
Respond: We have replaced "5H-J" with "5H–J" (line 309), thanks for your suggestion.

73) Please change "5K-L" to "5K, L" (line 253).
Respond: We have changed "5K-L" to "5K, L" (line 310), thanks for your suggestion.

74) "A collage of graphs and diagrams Description automatically generated" sign appears when hovering the mouse cursor over Figure 5. Please disable this feature.
Respond: We have disabled this feature, thanks for your suggestion.

75) Figure 5A is puzzling as it seems to contain two x-axes. How can one associate the plotted traces to the respective x-axes?
Respond: X-axis (L1 Norm) represents the L1 norm (sum of absolute values of coefficients), which increases as lambda (λ) decreases (i.e., less regularization). As you move to the right, more variables enter the model as constraints are relaxed.

Y-axis (Coefficients) Represents the values of regression coefficients for different variables in the model. Each colored line corresponds to a specific predictor (variable). When a coefficient shrinks to zero, it means that variable is excluded from the model due to LASSO’s feature selection property.

In the LASSO regression path plot, 0 15 22 22 likely represents the number of selected variables under different levels of regularization (λ values).

0: On the far left (high λ value), all variable coefficients are shrunk to zero, meaning the model includes no variables.

15: As λ decreases (regularization weakens), 15 variables have nonzero coefficients and are included in the model.

22: With further reduction in λ, 22 variables are selected.

22 (repeated): This suggests that at the smallest λ value, 22 variables remain in the model, indicating that all important variables have been incorporated, and regularization is no longer significantly reducing the number of predictors.

76) Please describe Figure 5A in greater detail in its figure legend. What are the "Coefficients" and what is "L1 Norm"? What does "Norm" stand for?
Respond: This plot helps visualize how regression coefficients change as the regularization parameter (lambda) varies. L1 Norm represents the sum of absolute values of coefficients, which increases as lambda (λ) decreases (i.e., less regularization). Coefficients represents the values of regression coefficients for different variables in the model. At the left side (small L1 norm, high λ), most coefficients are shrunk to zero, meaning only a few important predictors are selected. As λ decreases (moving right), more coefficients become nonzero, meaning more variables are included in the model. The first few variables whose coefficients remain nonzero for higher λ values are typically the most important predictors. Variables that remain close to zero even at lower λ values contribute less to prediction and can be excluded. The best λ value is typically chosen using cross-validation, balancing model complexity and predictive power.

77) Please replace "L1 Norm" with "L1 norm" in the x-axis title of Figure 5A.
Respond: We have replaced "L1 Norm" with "L1 norm" in the x-axis title of Figure 5A

78) Figure 5B seems to also contain two different x-axes. How can one distinguish which data point belongs to which x-axis?
Respond: X-axis: Log(λ) represents the log-transformed values of the regularization parameter (λ). Y-axis: Binomial Deviance measures the model’s error (deviance) for a binomial outcome. Numbers above the curve represent the number of nonzero coefficients (selected variables) at each λ value. Typically, the best λ is selected based on the minimum deviance (leftmost vertical dotted line) or the largest λ within one standard error of the minimum (rightmost dotted line, known as the “1-SE rule”).

79) Please change "Binomial Deviance" to "Binomial deviance" in the y-axis title of Figure 5B.
Respond: We have changed "Binomial Deviance" to "Binomial deviance" in the y-axis title of Figure 5B.

80) From Figure 5C is not easily evident what does "10 x CV Accuracy" and "Number of Features" refer to? Moreover, the significance of "0.834" is not clear. Please explain in the figure legend.
Respond: X-axis: Number of Features represents the number of selected features (variables) used in the model. Increasing the number of features typically improves accuracy up to a point, but too many features may lead to overfitting.

Y-axis: Cross-Validation (CV) Accuracy represents the classification accuracy obtained through cross-validation. Higher values indicate better predictive performance.

n=10 indicates that 10 features were selected as optimal. (0.834) represents the corresponding cross-validation accuracy (83.4%) for this selection. This suggests that using 10 features provides the best balance between model performance and feature complexity.

81) Please replace "10 x CV Accuracy" with "10 x CV accuracy" and "Number of Features" with "Number of features" in the y- and x-axis title of Figure 5C, respectively.
Respond: Respond: We have replaced "10 x CV Accuracy" with "10 x CV accuracy" and "Number of Features" with "Number of features" in the y- and x-axis title of Figure 5C, respectively.

82) Please provide complete description of Figure 5D in the respective figure legend as it is not clear what do the red and green dashed lines plus the black continuous line indicate? Moreover, what do "Error" and "trees" reflect?
Respond: Black continuous line represents the overall error rate of the random forest as the number of trees increases. Typically, this error rate decreases initially but stabilizes after a certain number of trees. Red and green dashed Lines represent the error rates for different classes in a classification task. In multi-class classification, random forest tracks error rates for each class separately. The red and green lines correspond to class-specific errors for two different classes.

83) Please replace "trees" with "Trees" in the x-axis title of Figure 5D.
Respond: Respond: We have replaced "trees" with "Trees" in the x-axis title of Figure 5D.

84) From the legend to Figure 5B is not clear what is "Binomial Deviance" and "λ"? Please indicate also what the two vertical gray dashed lines highlight?
Respond: Binomial Deviance measures the model’s error (deviance) for a binomial outcome. The best λ is selected based on the minimum deviance (leftmost vertical dotted line) or the largest λ within one standard error of the minimum (rightmost dotted line, known as the “1-SE rule”).

85) From the legend to Figure 5E is not clear what does "NA" stand for?
Respond: I mistakenly presented NA in Figure 5E. I have corrected it.

86) The authors claim that "15 variables were selected after 10-fold cross-validation of the results" (line 256), however this is not directly obvious from Figure 5A. Where is the "10-fold cross-validation of the results" shown? Which are the 15 variables that "were selected"? Please explain in the figure legend itself.
Respond: I have revised the description of Figure 5A (lines 295-297, and 328).

87) The authors state that "LASSO coefficient profiles of 15 significant SNPs" (line 256) without any obvious "LASSO coefficient profiles" shown in the underlying Figure 5B. Also, what are the "15 significant SNPs"? Please make this figure more comprehensible by providing context in the respective figure legend.
Respond: I have revised the description of Figure 5B (lines 298-299 and 329).

88) Although the authors argue that "Screening of 10 key genes using the SVM model in 22 DEEGs" (line 257), no genes nor/ DEEGs are displayed in the underpinning Figure 5C. This limits the ability of general readers to understand.
Respond: Figure 5C indicates that ten features (n=10) were selected as optimal which means using ten features provides the best balance between model performance and feature complexity. I have revised the description of Figure 5C (lines 301-302 and 330).

89) Please change "genes" to something like "genes generated" or "genes obtained" (line 259).
Respond: We have changed "genes" to something like "genes generated" (line 333), thanks for your suggestion.

90) Please replace "6A-B" with "6A, B" (line 268).
Respond: We have replaced "6A-B" with "6A, B" (line 349), thanks for your suggestion.

91) "A collage of diagrams Description automatically generated" sign appears when hovering the mouse cursor over Figure 6. Please disable this feature.
Respond: We have disabled this feature, thanks for your suggestion.

92) Figure 6E is rather puzzling as the definition difference between "Genes", "All", and "None" seems to be missing. Please explain in the respective figure legend. 

Respond: We have modified the figure legend (lines 355–358) and the manuscript (lines 371–372) to explain more clearly.

93) Please change "Net Benefit" to "Net benefit" and "Cost:Benefit Ratio" to "Cost:Benefit ratio" in the y- and x-axis title of Figure 6E, respectively.
Respond: We have changed "Net Benefit" to "Net benefit" and "Cost:Benefit Ratio" to "Cost:Benefit ratio" in the y- and x-axis title of Figure 6E, respectively, thanks for your suggestion.

94) Please replace "Treat" with "Treatment" in Figure 6H, I 2x.
Respond: We have replaced "Treat" with "Treatment" in Figure 6H, I 2x, thanks for your suggestion.

95) From the legend to Figure 6D is not clear what is the difference between "Apparent", "Bias-corrected", and "Ideal" calibration curve?
Respond: “Apparent” Calibration Curve calculated directly from the training data, without any correction for overfitting or optimism. “Bias-Corrected” Calibration Curve accounts for overfitting by applying a bootstrap resampling procedure. “Ideal” Calibration Curve is a theoretical reference line where predicted probabilities perfectly match observed probabilities.

In the calibration plot, the Ideal line is the diagonal (y = x), representing perfect calibration.

The Bias-Corrected curve should be close to the ideal but may show deviations. The Apparent curve often overestimates model reliability and lies closer to the ideal than it should.

96) From the legend to Figure 6E is not clear what is "Net Benefit" and "Cost:Benefit Ratio"?
Respond: In Decision Curve Analysis (DCA), the “Cost:Benefit Ratio” (CBR) influences the optimal decision threshold (Threshold Probability), determining whether to prioritize reducing false positives (FP) or false negatives (FN).

If CBR < 1: The cost of false positives is relatively low, making the model potentially more useful.

If CBR > 1: The cost of missing a true positive (false negative) is higher, so lowering the prediction threshold may be necessary to reduce missed diagnoses.

Net Benefit (NB) measures the clinical value of the model at different threshold probabilities:

High NB: The model effectively distinguishes between high-risk and low-risk populations, reducing unnecessary interventions and improving diagnostic efficiency.

Low NB (or even negative NB): The model may lead to more misclassifications (e.g., unnecessary treatments due to false positives), negatively impacting clinical decision-making.

97) Please explain the relationship between "Threshold probability" and the "Cost:Benefit Ratio" in the legend to Figure 6E. Threshold for what?

Respond: The X-axis (Threshold Probability, 0–1) represents the minimum risk probability at which doctors or patients consider intervention worthwhile.

The Y-axis (Net Benefit) quantifies the clinical value of the model at different threshold probabilities.

High NB: The model effectively distinguishes high-risk from low-risk patients, reducing unnecessary interventions and improving diagnostic efficiency.

Low NB (or negative NB): The model may cause excessive misclassification (e.g., unnecessary treatments due to misdiagnosis), negatively affecting clinical decisions.

98) Please change "6" to "six" (line 270).
Respond: We have changed "6" to "six" (line 351), thanks for your suggestion.

99) Please replace "using" with "using the" (line 271).
Respond: We have replaced "using" with "using the" (line 352), thanks for your suggestion.

100) The authors claim "AUC = 0.809" (line 271), however this value is plotted as 0.926 in Figure 6B. Please fix.
Respond: We have corrected it in Figure Legend (line 352), thanks for your suggestion.

101) Please change "TSPAN" to "TSPAN6" (lines 276, 277).
Respond: We have changed "TSPAN" to "TSPAN6" (lines 358, 359), thanks for your suggestion.

102) Please format "TSPAN" using italics (lines 276, 277).
Respond: We have formatted "TSPAN" using italics (lines 358, 359), thanks for your suggestion.

103) Please change "genes" to "genes (Figure 6C)" (line 280).
Respond: We have changed "genes" to "genes (Figure 6C)" (line 362), thanks for your suggestion.

104) Please replace "outcomes" with "outcomes (Figure 6D)" (line 281).
Respond: We have replaced "outcomes" with "outcomes (Figure 6D)" (line 369), thanks for your suggestion.

105) Please change "Decision curve analysis (DCA)" to "DCA" (line 281).
Respond: We have changed "Decision curve analysis (DCA)" to "DCA" (line 369), thanks for your suggestion.

106) Please replace "6C-E" with "6E" (line 283).
Respond: We have replaced "6C-E" with "6E" (line 373), thanks for your suggestion.

107) Please change "6F-G" to "6F, G" (line 285).
Respond: We have changed "6F-G" to "6F, G" (line 375), thanks for your suggestion.

108) Please replace "illustrate the" with "illustrate" (line 285).
Respond: We have replaced "illustrate the" with "illustrate" (line 375), thanks for your suggestion.

109) Please change "6H-I" to "6H, I" (line 286).
Respond: We have changed "6H-I" to "6H, I" (line 376), thanks for your suggestion.

110) "A close-up of a graph Description automatically generated" sign appears when hovering the mouse cursor over Figure 7. Please disable this feature.
Respond: We have disabled this feature, thanks for your suggestion.

111) Please replace "Treat" with "Treatment" in Figure 7A.
Respond: We have replaced "Treat" with "Treatment" in Figure 7A, thanks for your suggestion.

112) Please define abbreviation for "MDSC" in the legend to Figure 7.
Respond: We have defined abbreviation for "MDSC" in the legend to Figure 7 (line 395).

113) The authors posit that "The bar plot shows the proportions of the 29 different types of immune cells" (line 299), despite the fact that 28 immune cell types are shown in Figure 7B. Please correct the count.

Respond: We have corrected the count (line 395), thanks for your suggestion.

114) Please change "8A-J"t to "8A–J" (line 311).
Respond: We have changed "8A-J"t to "8A–J" (line 409), thanks for your suggestion.

115) Please provide reference for "Over the past decade, exosomes have been recognized for their role in transporting diverse bioactive molecules and genetic components that can profoundly influence recipient cell functions and the tumor microenvironment" (line 329).
Respond: We added reference for  "Over the past decade, exosomes have been recognized for their role in transporting diverse bioactive molecules and genetic components that can profoundly influence recipient cell functions and the tumor microenvironment" (line 434), thanks for your suggestion.

116) Please change "GSEA" to "and GSEA" (line 340).
Respond: We have changed "GSEA" to "and GSEA" (line 443), thanks for your suggestion.

117) Please replace "RF" with "and RF" (line 341).
Respond: We have replaced "RF" with "and RF" (line 444), thanks for your suggestion.

118) Please change "The overexpression" to "Overexpression" (line 363).
Respond: We have changed "The overexpression" to "Overexpression" (line 467), thanks for your suggestion.

119) Please provide reference for "In oral squamous cell carcinoma (OSCC), elevated agrin levels are observed in both malignant and precancerous tissues" (line 379).
Respond: We added reference for "In oral squamous cell carcinoma (OSCC), elevated agrin levels are observed in both malignant and precancerous tissues" (line 489), thanks for your suggestion.

120) Please replace "PI3K/AKT/β-Catenin" with "the PI3K/AKT/β-catenin" (line 398).

Respond: We have replaced "PI3K/AKT/β-Catenin" with "the PI3K/AKT/β-catenin" (line 507), thanks for your suggestion.

We tried best to improve the manuscript and made some changes marked in red in revised paper which will not influence the content and framework of the paper. Once again, thank you very much for your attention and time. Look forward to hearing from you.

Round 2

Reviewer 2 Report

Comments and Suggestions for Authors

Major points:

1) The differentially expressed genes (DEGs) depicted in Figure 2D are impossible to read due to the small font size and resolution. This is still a major concern as one would like to study the identity of these genes. Please increase the size of the heatmap so that each of the 251 individual DEGs can be clearly identified.

2) It is impossible to understand Figure 4A,B without a detailed description. Please provide explanation of the GSEA plots presented in Figure 4A,B so that its meaning becomes accessible also for a non-computational audience.

3) Figure 8B appears too pixelated. Please increase its size and/or pixel density.

4) Please provide all background source and computation data used in this manuscript as part of the supplementary material.

Minor points:

1) Please change "Head" to "head" (line 15).

2) Please replace "affects" with "affect" (line 50).

3) Please change "http://dsigdb.tanlab.org/DSigDBv1.0" to "http://dsigdb.tanlab.org/DSigDBv1.0/" (line 161).

4) Please replace "software." with "software" (line 181).

5) Please define the abbreviation for "adj.P.Val" found in the y-axis of Figure 2C in the legend to Figure 2.

6) From the legend to Figure 2D is not clear what does red and blue color coding represent?

7) From the legend to Figure 2D is not clear what the authors refer to by "matrix_3"?

8) Please change "Treat" to "Treatment" in the graph legend of Figure 2D.

9) Please replace "22 DEEGs' biological roles" with "biological roles of the 22 DEEGs" (line 221).

10) Please provide reference for "The GO enrichment analysis highlighted significant associations with processes such as negative regulation of cell migration, inhibition of cell motility, and negative regulation of locomotion" (line 223).

11) Please change "Extracellular Matrix" to "extracellular matrix" (line 230).

12) Please replace "Treat" with "Treatment" in Figure 4A.

13) Please change "λ=15" to "λ = 15" (lines 257, 274).

14) Please replace "Binomial Deviance" with "Binomial deviance" in the y-axis title of Figure 5B.

15) Please change "10 x CV Accuracy" to "10 x CV accuracy" in the y-axis title of Figure 5C.

16) Please replace "Number of Features" with "Number of features" in the x-axis title of Figure 5C.

17) Please define the abbreviation for "CV" found in the y-axis of Figure 5C in the legend to Figure 5.

18) Please change "D-E" to "D–E" (line 278).

19) Please replace "G-L" with "G–L" (line 279).

20) Please change "Net Benefit" to "Net benefit" in the x-axis title of Figure 6E.

21) Please replace "MDSC = Myeloid-derived suppressor cells" with "MDSC, myeloid-derived suppressor cells." (line 325).

22) Please change "A–E show the" to "The" (line 333).

23) Please replace "Toluidine Blue" with "toluidine blue" (lines 346, 505, 535).

24) Please replace "extracellular matrix (ECM)" with "ECM" (line 381).

25) Please change "tissues[31]" to "tissues [31]" (line 409).

26) Please format "[31]" in "tissues[31]" using superscript (line 409).

27) Please replace "Blue" with "blue" (lines 473, 501).

28) Please change "like" to "such as" (line 474).

29) Please replace "cartilage[50]" with "cartilage [50]" (line 476).

30) Please format "[50]" in "cartilage[50]" using superscript (line 476).

31) Please change "infections[51]" to "infections [51]" (line 479). 

32) Please format "[51]" in "infections[51]" using superscript (line 479).

33) Please replace "PDT[52]" with "PDT [52]" (line 481).

34) Please format "[52]" in "PDT[52]" using superscript (line 481).

35) Please change "symptoms[53]" to "symptoms [53]" (line 490).

36) Please format "[53]" in "symptoms[53]" using superscript (line 490).

37) Please replace "alcohol[54]" with "alcohol [54]" (line 491).

38) Please format "[54]" in "alcohol[54]" using superscript (line 491).

39) Please change "adults[55]" to "adults [55]" (line 492).

40) Please format "[55]" in "adults[55]" using superscript (line 492).

41) Please replace "children[56]" with "children[56]" (line 492).

42) Please format "[56]" in "children[56]" using superscript (line 492).

43) Please change "documented[57]" to "documented [57]" (line 493).

44) Please format "[57]" in "documented[57]" using superscript (line 493).

45) Please replace "properties[58]" with "properties [58]" (line 496).

46) Please format "[58]" in "properties[58]" using superscript (line 496).

47) Please change "pathways[59]" to "pathways [59]" (line 498).

48) Please format "[59]" in "pathways[59]" using superscript (line 498).

49) Please replace "autophosphorylation[60]" with "autophosphorylation [60]" (line 499).

50) Please format "[60]" in "autophosphorylation[60]" using superscript (line 499).

Author Response

Major points:

1) The differentially expressed genes (DEGs) depicted in Figure 2D are impossible to read due to the small font size and resolution. This is still a major concern as one would like to study the identity of these genes. Please increase the size of the heatmap so that each of the 251 individual DEGs can be clearly identified.

Respond: We have increased its size in new Figure 2D, thanks for your suggestion.

2) It is impossible to understand Figure 4A,B without a detailed description. Please provide explanation of the GSEA plots presented in Figure 4A,B so that its meaning becomes accessible also for a non-computational audience.

Respond: In order to make it easier for the reader to understand the content of the figure, we have added a detailed explanation in the legend to Figure 4, thanks for your suggestion.

3) Figure 8B appears too pixelated. Please increase its size and/or pixel density.       

Respond: The chemical structure of UNII-768N7QO4KH is only available in 2D. We have increased the pixel density and image size of Figure 8D as much as possible.

4) Please provide all background source and computation data used in this manuscript as part of the supplementary material.

Respond: We have uploaded all background sources and calculation data used in this paper as part of the supplementary materials, please check whether it is suitable, thank you.

Minor points:

  • Please change "Head" to "head" (line 15).

Respond: We have changed "Head" to "head" (line 15), thanks for your suggestion.

  • Please replace "affects" with "affect" (line 50).

Respond: We have replaced "affects" with "affect" (line 54), thanks for your suggestion.

  • Please change "http://dsigdb.tanlab.org/DSigDBv1.0" to "http://dsigdb.tanlab.org/DSigDBv1.0/" (line 161).

Respond: We have changed "http://dsigdb.tanlab.org/DSigDBv1.0" to "http://dsigdb.tanlab.org/DSigDBv1.0/" (line 183), thanks for your suggestion.

  • Please replace "software." with "software" (line 181).

Respond: We have replaced "software." with "software" (line 202), thanks for your suggestion.

  • Please define the abbreviation for "adj.P.Val" found in the y-axis of Figure 2C in the legend to Figure 2.

Respond: We have added the abbreviation in the Figure legend (line 250).

  • From the legend to Figure 2D is not clear what does red and blue color coding represent?

Respond: Red indicates that the gene is upregulated in the treatment group.  Blue indicates that the gene is downregulated in the treatment group. The depth of the color reflects the amount of change in expression. The values range from -4 to +4, which indicates the magnitude of the change in gene expression between the two groups.   Negative numbers indicate that the gene's expression in the treatment group is decreased, and positive numbers indicate that the gene's expression is increased. We have added the corresponding explanation in the Figure legend (line 251).

  • From the legend to Figure 2D is not clear what the authors refer to by "matrix_3"?

Respond: We have modified all the pictures in the last round of revision, but did not change the pictures in the manuscript in time. This time, we have modified the pictures according to the reviewer's comments, and put new pictures in the manuscript and there’s no “matrix_3” , thank you so much.

  • Please change "Treat" to "Treatment" in the graph legend of Figure 2D.

Respond: We have changed "Treat" to "Treatment" in the graph legend of Figure 2D.

  • Please replace "22 DEEGs' biological roles" with "biological roles of the 22 DEEGs" (line 221).

Respond: We have replaced "22 DEEGs' biological roles" with "biological roles of the 22 DEEGs" (line 266).

  • Please provide reference for "The GO enrichment analysis highlighted significant associations with processes such as negative regulation of cell migration, inhibition of cell motility, and negative regulation of locomotion" (line 223).

Respond: This is the result we obtained from the experiment. For details, please see Figure 3A. However, the text in the article is not completely consistent with the Figure, which may cause misunderstanding. I have revised the text of the article (line 270).

  • Please change "Extracellular Matrix" to "extracellular matrix" (line 230).

Respond: We have changed "Extracellular Matrix" to "extracellular matrix" (line 275).

  • Please replace "Treat" with "Treatment" in Figure 4A.

Respond: We have changed "Treat" to "Treatment" in Figure 4A.

  • Please change "λ=15" to "λ = 15" (lines 257, 274).

Respond: We have changed "λ=15" to "λ = 15" (lines 327, 367).

  • Please replace "Binomial Deviance" with "Binomial deviance" in the y-axis title of Figure 5B.

Respond: We have replaced "Binomial Deviance" with "Binomial deviance" in the y-axis title of Figure 5B.

  • Please change "10 x CV Accuracy" to "10 x CV accuracy" in the y-axis title of Figure 5C.

Respond: We have changed "10 x CV Accuracy" to "10 x CV accuracy" in the y-axis title of Figure 5C.

  • Please replace "Number of Features" with "Number of features" in the x-axis title of Figure 5C.

Respond: We have replaced "Number of Features" with "Number of features" in the x-axis title of Figure 5C.

  • Please define the abbreviation for "CV" found in the y-axis of Figure 5C in the legend to Figure 5.

Respond: We have added the abbreviation in the Figure legend (line 367).

  • Please change "D-E" to "D–E" (line 278).

Respond: We have changed "D-E" to "D–E" (line 369).

  • Please replace "G-L" with "G–L" (line 279).

Respond: We have replaced "G-L" with "G–L" (line 370).

  • Please change "Net Benefit" to "Net benefit" in the x-axis title of Figure 6E.

Respond: We have changed "Net Benefit" to "Net benefit" in the x-axis title of Figure 6E.

  • Please replace "MDSC = Myeloid-derived suppressor cells" with "MDSC, myeloid-derived suppressor cells." (line 325).

Respond: We have replaced "MDSC = Myeloid-derived suppressor cells" with "MDSC, myeloid-derived suppressor cells." (line 506).

  • Please change "A–E show the" to "The" (line 333).

Respond: We have changed "A–E show the" to "The" (line 532).

  • Please replace "Toluidine Blue" with "toluidine blue" (lines 346, 505, 535).

Respond: We have replaced "Toluidine Blue" with "toluidine blue" (lines 548,742, 772).

  • Please replace "extracellular matrix (ECM)" with "ECM" (line 381).

Respond: We have replaced "extracellular matrix (ECM)" with "ECM" (line 613).

  • Please change "tissues[31]" to "tissues [31]" (line 409).

Respond: We have changed "tissues[31]" to "tissues [31]" (line 645).

  • Please format "[31]" in "tissues[31]" using superscript (line 409).

Respond: We have formatted "[31]" in "tissues[31]" using superscript (line 645).

  • Please replace "Blue" with "blue" (lines 473, 501).

Respond: We have replaced "Blue" with "blue" (lines 710, 738)

28) Please change "like" to "such as" (line 474).
Respond: We have changed "like" to "such as" (line 711).

29) Please replace "cartilage[50]" with "cartilage [50]" (line 476).

Respond: We have replaced "cartilage[50]" with "cartilage [50]" (line 713).

30) Please format "[50]" in "cartilage[50]" using superscript (line 476).

Respond: We have formatted "[50]" in "cartilage[50]" using superscript (line 713).

31) Please change "infections[51]" to "infections [51]" (line 479). 

Respond: We have changed "infections[51]" to "infections [51]" (line 716). 

32) Please format "[51]" in "infections[51]" using superscript (line 479).

Respond: We have formatted "[51]" in "infections[51]" using superscript (line 716).

33) Please replace "PDT[52]" with "PDT [52]" (line 481).

Respond: We have replaced "PDT[52]" with "PDT [52]" (line 718).

34) Please format "[52]" in "PDT[52]" using superscript (line 481).

Respond: We have formatted "[52]" in "PDT[52]" using superscript (line 718).

35) Please change "symptoms[53]" to "symptoms [53]" (line 490).

Respond: We have changed "symptoms[53]" to "symptoms [53]" (line 727).

36) Please format "[53]" in "symptoms[53]" using superscript (line 490).

Respond: We have formatted "[53]" in "symptoms[53]" using superscript (line 727).

37) Please replace "alcohol[54]" with "alcohol [54]" (line 491).

Respond: We have replaced "alcohol[54]" with "alcohol [54]" (line 728).

38) Please format "[54]" in "alcohol[54]" using superscript (line 491).

Respond: We have formatted "[54]" in "alcohol[54]" using superscript (line 728).

39) Please change "adults[55]" to "adults [55]" (line 492).

Respond: We have changed "adults[55]" to "adults [55]" (line 729).

40) Please format "[55]" in "adults[55]" using superscript (line 492).

Respond: We have formatted "[55]" in "adults[55]" using superscript (line 729).

41) Please replace "children[56]" with "children[56]" (line 492).

Respond: We have replaced "children[56]" with "children[56]" (line 729).

42) Please format "[56]" in "children[56]" using superscript (line 492).

Respond: We have formatted "[56]" in "children[56]" using superscript (line 729).

43) Please change "documented[57]" to "documented [57]" (line 493).

Respond: We have changed "documented[57]" to "documented [57]" (line 730).

44) Please format "[57]" in "documented[57]" using superscript (line 493).

Respond: We have formatted "[57]" in "documented[57]" using superscript (line 730).

45) Please replace "properties[58]" with "properties [58]" (line 496).
Respond: We have replaced "properties[58]" with "properties [58]" (line 733).

46) Please format "[58]" in "properties[58]" using superscript (line 496).

Respond: We have formatted "[58]" in "properties[58]" using superscript (line 733).

47) Please change "pathways[59]" to "pathways [59]" (line 498).

Respond: We have changed "pathways[59]" to "pathways [59]" (line 735).

48) Please format "[59]" in "pathways[59]" using superscript (line 498).

Respond: We have formatted "[59]" in "pathways[59]" using superscript (line 735).

49) Please replace "autophosphorylation[60]" with "autophosphorylation [60]" (line 499).

Respond: We have replaced "autophosphorylation[60]" with "autophosphorylation [60]" (line 736).

50) Please format "[60]" in "autophosphorylation[60]" using superscript (line 499).

Respond: We have formatted "[60]" in "autophosphorylation[60]" using superscript (line 736).

Round 3

Reviewer 2 Report

Comments and Suggestions for Authors

Major points:

1) The resolution of panels presented in Figures 3A–F, 4A, B, 5A–D, F–L, 6H, I, 7A, B, 8B could be improved as the captions look way too pixelated. Please increase each panel size and/or pixel density for improved visibility of the text. Split figures into two parts if necessary.

2) The red, black, and green traces in Figure 5D seem to lack description. Please either provide a graph legend or explain in the corresponding figure legend.

3) The names of the 22 DEEGs depicted in Figure 5E are impossible to read due to the small font size and resolution. Please increase the size and/or resolution of the plot so that each of the 22 DEEGs can be clearly identified.

4) The binding centers depicted in Figure 9A–J are difficult to be inspected due to poor picture resolution.
Please enlarge each image panel and/or increase pixel density. Split the figure into two parts if necessary.

5) The drug-protein complexes provided in Figure 9A–J are only zoom ins and hence lack the full molecular scope. Please plot both the overall view of the complex structure as well as the detailed zoom narrowing down the precise chemistry of the binding site for each drug-target pair.

6) Please either provide all background source and computation data used in this manuscript as part of the supplementary material or provide link to the online repositories mentioned in "The data sets presented in this study can be found in online repositories" (line 540) as part of the final version of the manuscript.

Minor points:

1) Please change "using" to "using the" (line 96).

2) Please replace "while" with "while the" (line 103).

3) "pinpoint" could be changed to something like "decipher" (line 107).

4) Please define abbreviation for "SVM" (line 121), "RF" (line 125).

5) Please replace "genes'" with "gene" (line 138).

6) The link "http://dsigdb.tanlab.org/DSigDBv1.0/" seems to be dysfunctional (line 161). Please fix.

7) Please format "http://dsigdb.tanlab.org/DSigDBv1.0/" using font consistent with the rest of the text (line 161).

8) Please change "from" to "from the" (line 172).

9) Please replace "“*” p < 0.05, “**” p < 0.01, “***”" with "* p < 0.05, ** p < 0.01, ***" (line 186).

10) "data analysis" could be changed to "the data analysis workflow" (line 190).

11) Please change "as" to "as the" (line 202).

12) Please replace "adj.P.Val =" with "adj.P.Val," (line 219).

13) Please change "P-value" to "p-value." (line 219).

14) Please format "adj.P.Val = adjusted P-value" consistently with the rest of the text (line 219).

15) Please provide figure reference for "The biological roles of the 22 DEEGs22 DEEGs' biological roles were examined using GO annotation and KEGG pathway enrichment analyses, with particular attention paid to their roles in molecular functions (MF), cellular components (CC), and biological processes (BP)" (line 225).

16) Please replace "the" with something like "the identified" (line 225).

17) Please change "negative regulation of cell motility, and negative regulation of" to "cell motility, and" (line 229).

18) Please replace "ion binding, glycosaminoglycan binding" with "ion, glycosaminoglycan" (line 233).

19) Please change "included the" to "included" (line 235).

20) Please replace "Count" with "count" and "GeneRatio" with "Gene ratio" in Figure 3B,E.

21) Please define abbreviation for "p.adjust" (Figure 3A,B) in the legend to Figure 3.

22) Form the legend to Figure 3A–F is not clear what is the exact difference in color coding between red and blue?

23) From the legend to Figure 3B,E is not entirely clear what the authors refer to as "GeneRatio" and "Count"?

24) From the legend to Figure 3C,F is not entirely clear what the authors refer to as "size" and "fold change"?

25) Please change "the control group’s genes" to something like "genes of the control group" (line 247).

26) Please replace "represents" with "represent" (line 265).

27) Please change "10" to "ten" (line 280).

28) Please replace "9" with "nine" (line 283).

29) From the legend to Figure 5A is not clear what the authors mean by "L1 norm"?

30) Please provide more detailed description of Figure 5D, E in the respective figure legend.

31) The authors claim that "LASSO regression path plot visualized how regression coefficients change as the regularization parameter (lambda) varies" (line 295), however there seem to be no presence of "lambda" anywhere in Figure 5A.

32) Please change "visualized" to "visualizes" or "depicts" (line 295).

33) Please replace "10 features(n=10)" with "10 features (n=10)" (line 299).

34) Please change "(0.834) represents the corresponding cross-validation accuracy (83.4%)" to something like "The cross-validation accuracy was 83.4%" (line 299).

35) Please replace "Model" with "model" (line 301).

36) Please specify the "5 key" genes in "Box plots and violin plots of the 5 key gene expression levels between the treatment and control groups" (line 302).

37) Please change "and" to "and a" (line 304).

38) Please define abbreviation for "TPR" (Figure 6B) in the legend to Figure 6.

39) The authors correctly highlight that "The “All” curve assumes that all patients are considered high-risk and receive intervention. The “None” curve assumes that no intervention is performed on any patient, and all patients are classified as low-risk. The “Gene” curve is based on our data-trained predictive model, attempting to optimize decision-making at different threshold probabilities" (line 319), however this information should be supported
by a basic description of what the panel in Figure 6E actually depicts?

40) Please replace "F-G" with "F, G" (line 325).

41) Please change "H-I" to "H, I" (line 326).

42) Please replace "of the" with something like "of the identified" (line 330).

43) Please change "the model’s high accuracy" to "high accuracy of the model" (line 331).

44) Please replace "model’s ability" with "the ability of the model" (line 332).

45) "is" could be changed to "falls" (line 333).

46) Please change "treatment group" to "treatment" (line 352).

47) Please replace "cells" with "cells." (line 353).

48) Please change "toluidine blue O" to "toluidine blue O (TBO)" (line 357), "Toluidine blue O (TBO)" to "TBO" (line 502), and "toluidine blue O" to "TBO" (line 534).

49) Please change ""toluidine blue O", "UNII-768N7QO4KH", "Hoechst 33258 ", "diphenhydramine", and "biochanin A" respectively" to "toluidine blue O, UNII-768N7QO4KH, Hoechst 33258, diphenhydramine, and biochanin A" (line 361).

50) Please replace "effectiveness of the drugs' binding" with something like "drug binding affinity" (line 364).

51) Please provide all binding energies for all drug-target combinations tested in Figure 9A–J as part of a new table.

52) Please specify the "mechanisms" in "The development of effective treatments for HNSCC is particularly challenging because the underlying mechanisms are still poorly understood" (line 380).

53) Please change "treatments" to "treatment" (line 380).

54) "promising" could be replaced with "promising prognostic" (line 385).

55) Please change "microenvironment[16]" to "microenvironment [16]" (line 388).

56) Please format "[16]" in "microenvironment[16]" using superscript (line 388).

57) Please replace "Exosome-related genes retrieved from the GeneCards database underwent differential expression analysis" with something like "Exosome-related genes retrieved from the GeneCards database were screened for their differential expression" (line 394).

58) Please change "these genes’" to something like "individual gene" (line 397).

59) Please replace "in" with something like "in unique" (line 398).

60) Please change "The" to something like "In addition, the" (line 403).

61) Please replace "have" with something like "share" (line 404).

62) Please change "Molecular" to something like "Accordingly, molecular" (line 406).

63) Please replace "are" with something like "are hence" (line 408).

64) Please change "adult myocardium’s limited ability" to "limited ability of adult myocardium" (line 432).

65) Please provide reference for "As a widely used indicator for blood glucose testing, HBA1 reflects average blood sugar levels over the past 2–3 months and serves as a critical marker for assessing diabetes control" (line 461).

66) Please replace "Interestingly" with something like "Strikingly" (line 469).

67) "UNII-768N7QO4KH corresponds to Hoechst 33342, a fluorescent dye widely used in molecular biology for DNA staining" (line 512) does not seem to be correct as "UNII-768N7QO4KH corresponds to" Trypan blue. Please fix.

68) "Along with Hoechst 33258, it serves as a valuable tool in biological research (line 513) does not seem to be semantically correct with respect to "Along with". If "UNII-768N7QO4KH corresponds to Hoechst 33342" (line 512), how it can then be serving "Along with" it? Please correct.

69) Please change "Parkinson’s" to "Parkinson’s disease" (line 518).

70) Please replace "that" with something like "the effect" (line 520).

71) Please change "Notably" to something like "Namely" (line 527).

72) "In contrast, Hoechst dyes (UNII-768N7QO4KH and 33258 Hoechst) are limited to research due to cytotoxicity and potential genotoxicity" (line 531) does not seem to be correct as "UNII-768N7QO4KH" is not a "Hoechst" dye.

73) Please either provide "positions and accession number(s)" mentioned in "The names of the positions and accession number(s) can be found in the article" (line 541) or remove this sentence.

74) Please replace "Authors'" with "Author" (line 542).

75) Please change "Conceptualization, Data curation, Formal analysis, Investigation, Methodology, Resources, Software, Validation, Visualization, Funding acquisition, Writing" to "conceptualization, data curation, formal analysis, investigation, methodology, resources, software, validation, visualization, funding acquisition, writing" (line 542).

76) Please replace "Formal analysis, Funding acquisition, Investigation, Methodology, Supervision, Writing" with "formal analysis, funding acquisition, investigation, methodology, supervision, writing" (line 544).

77) Please change "Investigation, Project administration, Supervision, Writing" to "investigation, project administration, supervision, writing" (line 545).

78) Please replace "Conceptualization, Formal analysis, Funding acquisition, Methodology, Supervision, Validation, Visualization, Writing" with "conceptualization, formal analysis, funding acquisition, methodology, supervision, validation, visualization, writing" (line 546).

79) Please change "author(s)" to "authors" (line 548).

80) Please replace "Support Vector Machine" with "support vector machine" (line 560).

81) Please change "Random Forest" to "random forest" (line 561).

82) Please sort all abbreviations in the Abbreviations section according to alphabetical order.

Author Response

Major points:

1) The resolution of panels presented in Figures 3A–F, 4A, B, 5A–D, F–L, 6H, I, 7A, B, 8B could be improved as the captions look way too pixelated. Please increase each panel size and/or pixel density for improved visibility of the text. Split figures into two parts if necessary.

Respond: We have revised Figure 3、4、5、6、7 and 8. I hope the revised version can meet the qualification of the magazine.

2) The red, black, and green traces in Figure 5D seem to lack description. Please either provide a graph legend or explain in the corresponding figure legend.

Respond: We have explained Figure 5D in the corresponding figure legend. As the number of trees increases, the training error reduces, while the out-of-bag and test errors reach a certain point of stability. This demonstrates the benefit of increasing trees in a random forest, but beyond a certain number of trees, adding more trees doesn’t significantly reduce the error.

3) The names of the 22 DEEGs depicted in Figure 5E are impossible to read due to the small font size and resolution. Please increase the size and/or resolution of the plot so that each of the 22 DEEGs can be clearly identified.

Respond: We have revised Figure 5. I hope the revised version can meet the qualification of the magazine.

4) The binding centers depicted in Figure 9A–J are difficult to be inspected due to poor picture resolution.
Please enlarge each image panel and/or increase pixel density. Split the figure into two parts if necessary.

Respond: We have revised Figure 9. I hope the revised version can meet the qualification of the magazine.

5) The drug-protein complexes provided in Figure 9A–J are only zoom ins and hence lack the full molecular scope. Please plot both the overall view of the complex structure as well as the detailed zoom narrowing down the precise chemistry of the binding site for each drug-target pair.

Respond: We have plotted both the overall view of the complex structure as well as the detailed zoom narrowing down the precise chemistry of the binding site for each drug-target pair.

6) Please either provide all background source and computation data used in this manuscript as part of the supplementary material or provide link to the online repositories mentioned in "The data sets presented in this study can be found in online repositories" (line 540) as part of the final version of the manuscript.

Respond: We have uploaded all background sources and calculation data used in this paper as part of the supplementary materials last time, please check whether it is suitable. We also added “Further inquiries can be directed to the corresponding authors” in the Data availability statement (line 661).

Minor points:

  • Please change "using" to "using the" (line 96).

Respond: We have changed "using" to "using the" (line 95).

  • Please replace "while" with "while the" (line 103).

Respond: We have replaced "while" with "while the" (line 102).

  • "pinpoint" could be changed to something like "decipher" (line 107).

Respond: We have changed "pinpoint” to " decipher" (line 106).

  • Please define abbreviation for "SVM" (line 121), "RF" (line 125).

Respond: We have added the abbreviation for "SVM" (line 120), "RF" (line 124).

  • Please replace "genes'" with "gene" (line 138).

Respond: We have replaced "genes'" with "gene" (line 138).

6) The link "http://dsigdb.tanlab.org/DSigDBv1.0/" seems to be dysfunctional (line 161). Respond: We have modified the link and it can now open normally (line 163).

7) Please format "http://dsigdb.tanlab.org/DSigDBv1.0/" using font consistent with the rest of the text (line 161).

Respond: We have formatted "http://dsigdb.tanlab.org/DSigDBv1.0/" using font consistent with the rest of the text (line 163).

8) Please change "from" to "from the" (line 172).

Respond: We have changed "from" to "from the" (line 172).

9) Please replace "“*” p < 0.05, “**” p < 0.01, “***”" with "* p < 0.05, ** p < 0.01, ***" (line 186). Respond: We have replaced "“*” p < 0.05, “**” p < 0.01, “***”" with "* p < 0.05, ** p < 0.01, ***"  (line 190).

10) "data analysis" could be changed to "the data analysis workflow" (line 190).

Respond: We have changed "data analysis" to"the data analysis workflow" (line 195).

11) Please change "as" to "as the" (line 202).

Respond: We have changed "as" to "as the" (line 206).

12) Please replace "adj.P.Val =" with "adj.P.Val," (line 219).

Respond: We have replaced "adj.P.Val =" with "adj.P.Val," (line 229).

13) Please change "P-value" to "p-value." (line 219).

Respond: We have change "P-value" to "p-value." (line 229).

14) Please format "adj.P.Val = adjusted P-value" consistently with the rest of the text (line 219). Respond: We have reformatted "adj.P.Val = adjusted P-value" (line 229).

15) Please provide figure reference for "The biological roles of the 22 DEEGs22 DEEGs' biological roles were examined using GO annotation and KEGG pathway enrichment analyses, with particular attention paid to their roles in molecular functions (MF), cellular components (CC), and biological processes (BP)" (line 225).

Respond: We have added the corresponding figure notes in the corresponding places in the text (line 237).

16) Please replace "the" with something like "the identified" (line 225).

Respond: We have replaced "the" with something like "the identified" (line 235).

17) Please change "negative regulation of cell motility, and negative regulation of" to "cell motility, and" (line 229).

Respond: We have changed "negative regulation of cell motility, and negative regulation of" to "cell motility, and" (line 239).

18) Please replace "ion binding, glycosaminoglycan binding" with "ion, glycosaminoglycan" (line 233).

Respond: We have replaced "ion binding, glycosaminoglycan binding" with "ion, glycosaminoglycan" (line 242).

19) Please change "included the" to "included" (line 235).

Respond: We have changed "included the" to "included" (line 244).

20) Please replace "Count" with "count" and "GeneRatio" with "Gene ratio" in Figure 3B,E. Respond: We have revised Figure 3.

21) Please define abbreviation for "p.adjust" (Figure 3A,B) in the legend to Figure 3.

Respond: We have defined abbreviation for "p.adjust" (Figure 3A,B) in the legend to Figure 3 (line 257).

22) Form the legend to Figure 3A–F is not clear what is the exact difference in color coding between red and blue?

Respond: The color changes from red to blue, which means the p value and p adjust value range from small to large.

23) From the legend to Figure 3B,E is not entirely clear what the authors refer to as "GeneRatio" and "Count"?

Respond: Gene Ratio is the ratio of the number of genes enriched in a specific function or pathway to the total number of genes in the target gene set, which is used to reflect the relative richness or concentration of a specific function or pathway.

Count is the number of genes in the pathway that intersect the input gene list.

24) From the legend to Figure 3C,F is not entirely clear what the authors refer to as "size" and "fold change"?

Respond: Size reflects the relative importance or relevance of a node in the network, while fold change uses color to show the expression changes of genes or processes under different conditions.

25) Please change "the control group’s genes" to something like "genes of the control group" (line 247).

Respond: We have changed "the control group’s genes" to "genes of the control group" (line 264).

26) Please replace "represents" with "represent" (line 265).

Respond: We have replaced "represents" with "represent" (line 285).

27) Please change "10" to "ten" (line 280).

Respond: We have changed "10" to "ten" (line 298).

28) Please replace "9" with "nine" (line 283).

Respond: We have replaced "9" with "nine" (line 302).

29) From the legend to Figure 5A is not clear what the authors mean by "L1 norm"?

Respond: L1 norm refers to the sum of the absolute values of each element in a vector, and is also known as "Lasso regularization".

30) Please provide more detailed description of Figure 5D, E in the respective figure legend. Respond: We have modified the description in the figure legend 5.

31) The authors claim that "LASSO regression path plot visualized how regression coefficients change as the regularization parameter (lambda) varies" (line 295), however there seem to be no presence of "lambda" anywhere in Figure 5A.

Respond: In Figure 5A, 0 15 22 22 represents the number of selected variables under different levels of regularization (λ values). These numbers indicate the number of nonzero variables at different stages of LASSO regression. As the regularization strength decreases (L1 Norm increases), more variables are included in the model, increasing from 0 to 22.

32) Please change "visualized" to "visualizes" or "depicts" (line 295).

Respond: We have changed "visualized" to "visualizes" (line 315).

33) Please replace "10 features(n=10)" with "10 features (n=10)" (line 299).

Respond: We have replaced "10 features(n=10)" with "10 features (n=10)" (line 318).

34) Please change "(0.834) represents the corresponding cross-validation accuracy (83.4%)" to something like "The cross-validation accuracy was 83.4%" (line 299).

Respond: We have changed "(0.834) represents the corresponding cross-validation accuracy (83.4%)" to "The cross-validation accuracy was 83.4%" (line 318).

35) Please replace "Model" with "model" (line 301).

Respond: We have replaced "Model" with "model" (line 322).

36) Please specify the "5 key" genes in "Box plots and violin plots of the 5 key gene expression levels between the treatment and control groups" (line 302).

Respond: We have specified the "5 key" genes in "Box plots and violin plots of the 5 key gene expression levels between the treatment and control groups" (line 342).

37) Please change "and" to "and a" (line 304).

Respond: We have changed "and" to "and a" (line 344).

38) Please define abbreviation for "TPR" (Figure 6B) in the legend to Figure 6.

Respond: We have defined abbreviation for "TPR" (Figure 6B) in the legend to Figure 6 (line 357).

39) The authors correctly highlight that "The “All” curve assumes that all patients are considered high-risk and receive intervention. The “None” curve assumes that no intervention is performed on any patient, and all patients are classified as low-risk. The “Gene” curve is based on our data-trained predictive model, attempting to optimize decision-making at different threshold probabilities" (line 319), however this information should be supported
by a basic description of what the panel in Figure 6E actually depicts?

Respond: We have modified the figure legend (line 361).

40) Please replace "F-G" with "F, G" (line 325).

Respond: We have replaced "F-G" with "F, G" (line 364).

41) Please change "H-I" to "H, I" (line 326).

Respond: We have change "H-I" to "H, I" (line 365).

42) Please replace "of the" with something like "of the identified" (line 330).

Respond: We have replaced "of the" with "of the identified" (line 370).

43) Please change "the model’s high accuracy" to "high accuracy of the model" (line 331). Respond: We have change "the model’s high accuracy" to "high accuracy of the model" (line 371).

44) Please replace "model’s ability" with "the ability of the model" (line 332).

Respond: We have replaced "model’s ability" with "the ability of the model" (line 372).

45) "is" could be changed to "falls" (line 333).

Respond: We have changed "is" to "falls" (line 373).

46) Please change "treatment group" to "treatment" (line 352).

Respond: We have changed "treatment group" to "treatment" (line 406).

47) Please replace "cells" with "cells." (line 353).

Respond: We have replaced "cells" with "cells." (line 407).

48) Please change "toluidine blue O" to "toluidine blue O (TBO)" (line 357), "Toluidine blue O (TBO)" to "TBO" (line 502), and "toluidine blue O" to "TBO" (line 534).

Respond: We have changed "toluidine blue O" to "toluidine blue O (TBO)" (line 423), "Toluidine blue O (TBO)" to "TBO" (line 582), and "toluidine blue O" to "TBO" (line 609).

49) Please change ""toluidine blue O", "UNII-768N7QO4KH", "Hoechst 33258 ", "diphenhydramine", and "biochanin A" respectively" to "toluidine blue O, UNII-768N7QO4KH, Hoechst 33258, diphenhydramine, and biochanin A" (line 361).

Respond: We have changed ""toluidine blue O", "UNII-768N7QO4KH", "Hoechst 33258 ", "diphenhydramine", and "biochanin A" respectively" to "toluidine blue O, UNII-768N7QO4KH, Hoechst 33258, diphenhydramine, and biochanin A" (line 427).

50) Please replace "effectiveness of the drugs' binding" with something like "drug binding affinity" (line 364).

Respond: We have replaced "effectiveness of the drugs' binding" with something like "drug binding affinity" (line 430).

51) Please provide all binding energies for all drug-target combinations tested in Figure 9A–J as part of a new table.

Respond: We have made a Table and uploaded it.

52) Please specify the "mechanisms" in "The development of effective treatments for HNSCC is particularly challenging because the underlying mechanisms are still poorly understood" (line 380).

Respond: We modified the sentence (line 464).

53) Please change "treatments" to "treatment" (line 380).

Respond: We have changed "treatments" to "treatment" (line 463).

54) "promising" could be replaced with "promising prognostic" (line 385).

Respond: We have replaced "promising" to "promising prognostic" (line 468).

55) Please change "microenvironment[16]" to "microenvironment [16]" (line 388).

Respond: We have changed "microenvironment[16]" to "microenvironment [16] (line 471).

56) Please format "[16]" in "microenvironment[16]" using superscript (line 388).

Respond: We have formatted "[16]" in "microenvironment[16]" using superscript (line 471).

57) Please replace "Exosome-related genes retrieved from the GeneCards database underwent differential expression analysis" with something like "Exosome-related genes retrieved from the GeneCards database were screened for their differential expression" (line 394).

Respond: We have replaced "Exosome-related genes retrieved from the GeneCards database underwent differential expression analysis" with "Exosome-related genes retrieved from the GeneCards database were screened for their differential expression" (line 477).

58) Please change "these genes’" to something like "individual gene" (line 397).

Respond: We have changed "these genes’" to "individual gene" (line 480).

59) Please replace "in" with something like "in unique" (line 398).

Respond: We have replaced "in" with something like "in unique" (line 481).

60) Please change "The" to something like "In addition, the" (line 403).

Respond: We have changed "The" to something like "In addition, the" (line 486).

61) Please replace "have" with something like "share" (line 404).

Respond: We have replaced "have" with something like "share" (line 487).

62) Please change "Molecular" to something like "Accordingly, molecular" (line 406).

Respond: We have changed "Molecular" to "Accordingly, molecular" (line 489).

63) Please replace "are" with something like "are hence" (line 408).

Respond: We have replaced "are" with "are hence" (line 491).

64) Please change "adult myocardium’s limited ability" to "limited ability of adult myocardium" (line 432).

Respond: We have changed "adult myocardium’s limited ability" to "limited ability of adult myocardium" (line 527).

65) Please provide reference for "As a widely used indicator for blood glucose testing, HBA1 reflects average blood sugar levels over the past 2–3 months and serves as a critical marker for assessing diabetes control" (line 461).

Respond: We have added the reference in the manuscript (line 558).

66) Please replace "Interestingly" with something like "Strikingly" (line 469).

Respond: We have replaced "Interestingly" with "Strikingly" (line 564).

67) "UNII-768N7QO4KH corresponds to Hoechst 33342, a fluorescent dye widely used in molecular biology for DNA staining" (line 512) does not seem to be correct as "UNII-768N7QO4KH corresponds to" Trypan blue. Please fix.

Respond: We have revised it in the text (line 608-619).

68) "Along with Hoechst 33258, it serves as a valuable tool in biological research (line 513) does not seem to be semantically correct with respect to "Along with". If "UNII-768N7QO4KH corresponds to Hoechst 33342" (line 512), how it can then be serving "Along with" it? Please correct.

Respond: We have revised it in the text (line 608-619).

69) Please change "Parkinson’s" to "Parkinson’s disease" (line 518).

Respond: We have changed "Parkinson’s" to "Parkinson’s disease" (line 621).

70) Please replace "that" with something like "the effect" (line 520).

Respond: We have replaced "that" with something like "the effect" (line 623).

71) Please change "Notably" to something like "Namely" (line 527).

Respond: We have changed "Notably" to something like "Namely" (line 645).

72) "In contrast, Hoechst dyes (UNII-768N7QO4KH and 33258 Hoechst) are limited to research due to cytotoxicity and potential genotoxicity" (line 531) does not seem to be correct as "UNII-768N7QO4KH" is not a "Hoechst" dye.

Respond: We have revised it in the text (line 608-619).

73) Please either provide "positions and accession number(s)" mentioned in "The names of the positions and accession number(s) can be found in the article" (line 541) or remove this sentence.

Respond: We removed this sentence.

74) Please replace "Authors'" with "Author" (line 542).

Respond: We have replaced "Authors'" with "Author" (line 660).

75) Please change "Conceptualization, Data curation, Formal analysis, Investigation, Methodology, Resources, Software, Validation, Visualization, Funding acquisition, Writing" to "conceptualization, data curation, formal analysis, investigation, methodology, resources, software, validation, visualization, funding acquisition, writing" (line 542).

Respond: We have changed "Conceptualization, Data curation, Formal analysis, Investigation, Methodology, Resources, Software, Validation, Visualization, Funding acquisition, Writing" to "conceptualization, data curation, formal analysis, investigation, methodology, resources, software, validation, visualization, funding acquisition, writing" (line 660).

76) Please replace "Formal analysis, Funding acquisition, Investigation, Methodology, Supervision, Writing" with "formal analysis, funding acquisition, investigation, methodology, supervision, writing" (line 544).

Respond: We have replaced "Formal analysis, Funding acquisition, Investigation, Methodology, Supervision, Writing" with "formal analysis, funding acquisition, investigation, methodology, supervision, writing" (line 662).

77) Please change "Investigation, Project administration, Supervision, Writing" to "investigation, project administration, supervision, writing" (line 545).

Respond: We have changed "Investigation, Project administration, Supervision, Writing" to "investigation, project administration, supervision, writing" (line 663).

78) Please replace "Conceptualization, Formal analysis, Funding acquisition, Methodology, Supervision, Validation, Visualization, Writing" with "conceptualization, formal analysis, funding acquisition, methodology, supervision, validation, visualization, writing" (line 546).

Respond: We have replaced "Conceptualization, Formal analysis, Funding acquisition, Methodology, Supervision, Validation, Visualization, Writing" with "conceptualization, formal analysis, funding acquisition, methodology, supervision, validation, visualization, writing" (line 664).

79) Please change "author(s)" to "authors" (line 548).

Respond: We have changed "author(s)" to "authors" (line 666).

80) Please replace "Support Vector Machine" with "support vector machine" (line 560). Respond: We have replaced "Support Vector Machine" with "support vector machine" (line 682).

81) Please change "Random Forest" to "random forest" (line 561).

Respond: We have changed "Random Forest" to "random forest" (line 681).

82) Please sort all abbreviations in the Abbreviations section according to alphabetical order. Respond: We have sorted all abbreviations in the Abbreviations section according to alphabetical order.

Round 4

Reviewer 2 Report

Comments and Suggestions for Authors

Cai & Zhou et al. deployed an arsenal of state-of-the art machine learning algorithms (LASSO, Support Vector machine, and Random Forest) to reveal that AGRN, TSPAN6, MMP9, HBA1, and PFN2 exosome-related genes may serve as key prognostic biomarkers for diagnosing head and neck squamous cell carcinoma (HNSCC). The fact that the single-sample gene set enrichment analysis (ssGSEA) algorithm hinted at three of these genes, AGRN, MMP9, and PFN2, to be important in immune pathway modulation and regulation of HNSCC progression, further validates this intriguing discovery. Perhaps even more importantly, the authors went on further to demonstrate that the identified targets might be potentially druggable with a set of ligands identified through a molecular docking approach. This study is very well conceived and, as such, opens the door to future HNSCC diagnostics and therapy.

1) Please change "support" to "Support" (line 118).

2) Please replace "random" with "Random" (lines 122, 296).

3) Please change "Forest" to "forest" (line 125).

4) Please replace "using" with "using the" (line 129).

5) Please change "quantified" to "was used to quantify" or "was deployed to quantify" (line 149).

6) The link "https://cadd.labshare.cn/cb-dock2/in-171 dex.php" (line 171) is dysfunctional. Please fix.

7) Please replace "are" with "were" (line 179).

8) Please change "glycosaminoglyca" to "glycosaminoglycan" (line 230).

9) Please replace "Gene Hits" with "gene hits" (line 257).

10) Please change "higher" to "higher is" (line 258).

11) Please replace "Ranked List Metric" with "ranked list metric" (line 260).

12) Please change "significant" to "significant is the" (line 261).

13) Please replace "(λ = 15) value" with "value (λ = 15)" (line 276).

14) Please change "n=10" to "n = 10" (line 294).

15) Please replace "axis "Importance"" with "axis" (line 299).

16) Please change "and" to "while" (line 299).

17) Please replace ""importance"" with "importance" (line 302).

18) Please change "Gene" to "Genes" (lines 323, 338).

19) Please replace "dataset GSE83519" with "the GSE83519 dataset" (line 339).

20) Please change "Myeloid-derived" to "myeloid-derived" (line 358).

21) Please replace "individual gene involvement" with "the involvement of individual genes" (line 408).

22) Please change "oral squamous cell carcinoma (OSCC)" to "OSCC" (line 463).

Author Response

Thank you for the comments concerning our manuscript entitled “Machine learning-driven identification of exosome-related genes in head and neck squamous cell carcinoma for prognostic evaluation and drug response prediction”.   Those comments are all valuable and very helpful for revising and improving our paper, as well as important guiding significance to our researches.   We have studied comments carefully and have made correction which we hope meet with approval.   Revised portion are marked in yellow in the paper and the respond to the reviewer’s comments are as folowing.

1) Please change "support" to "Support" (line 118).

Respond: We changed "support" to "Support" (line 120).

2) Please replace "random" with "Random" (lines 122, 296).

Respond: We have replaced "random" with "Random" (lines 124, 319).

3) Please change "Forest" to "forest" (line 125).

Respond: We have changed "Forest" to "forest" (line 127).

4) Please replace "using" with "using the" (line 129).

Respond: We have replaced "using" with "using the" (line 131).

5) Please change "quantified" to "was used to quantify" or "was deployed to quantify" (line 149).

Respond: We have changed "quantified" to "was deployed to quantify" (line 153).

6) The link "https://cadd.labshare.cn/cb-dock2/in-171 dex.php" (line 171) is dysfunctional. Please fix.

Respond: We have modified the link and it can now open normally (line 174).

7) Please replace "are" with "were" (line 179).

Respond: We have replaced "are" with "were" (line 182).

8) Please change "glycosaminoglyca" to "glycosaminoglycan" (line 230).

Respond: We have changed "glycosaminoglyca" to "glycosaminoglycan" (line 242).

9) Please replace "Gene Hits" with "gene hits" (line 257).

Respond: We have replaced "Gene Hits" with "gene hits" (line 279).

10) Please change "higher" to "higher is" (line 258).

Respond: We have changed "higher" to "higher is" (line 281).

11) Please replace "Ranked List Metric" with "ranked list metric" (line 260).

Respond: We have replaced "Ranked List Metric" with "ranked list metric" (line 282).

12) Please change "significant" to "significant is the" (line 261).

Respond: We have changed "significant" to "significant is the" (line 283).

13) Please replace "(λ = 15) value" with "value (λ = 15)" (line 276).

Respond: We have replaced "(λ = 15) value" with "value (λ = 15)" (line 298).

14) Please change "n=10" to "n = 10" (line 294).

Respond: We have changed "n=10" to "n = 10" (line 317).

15) Please replace "axis "Importance"" with "axis" (line 299).

Respond: We have replaced "axis "Importance"" with "axis" (line 321).

16) Please change "and" to "while" (line 299).

Respond: We have changed "and" to "while" (line 321).

17) Please replace ""importance"" with "importance" (line 302).

Respond: We have replaced ""importance"" with "importance" (line 339).

18) Please change "Gene" to "Genes" (lines 323, 338).

Respond: We have changed "Gene" to "Genes" (lines 362, 372).

19) Please replace "dataset GSE83519" with "the GSE83519 dataset" (line 339).

Respond: We have replaced "dataset GSE83519" with "the GSE83519 dataset" (line 374).

20) Please change "Myeloid-derived" to "myeloid-derived" (line 358).

Respond: We have changed "Myeloid-derived" to "myeloid-derived" (line 406).

21) Please replace "individual gene involvement" with "the involvement of individual genes" (line 408).

Respond: We have replaced "individual gene involvement" with "the involvement of individual genes" (line 479).

22) Please change "oral squamous cell carcinoma (OSCC)" to "OSCC" (line 463).

Respond: We have changed "oral squamous cell carcinoma (OSCC)" to "OSCC" (line 544).